



# 1 OH and HO₂ radical chemistry in a midlatitude forest: Measurements and

# 2 model comparisons

Michelle M. Lew[1*], Pamela S. Rickly[2**], Brandon P. Bottorff[1], Sofia Sklaveniti[2,3], Thierry Léonardis[3],
Nadine Locoge[3], Sebastien Dusanter[3], Shuvashish Kundu[4***], Ezra Wood[5], and Philip S. Stevens[1,2]
[1] Department of Chemistry, Indiana University, Bloomington, IN 47405, USA
[2] O'Neill School of Public and Environmental Affairs, Indiana University, Bloomington, IN 47405, USA
[3] IMT Lille Douai, Univ. Lille, SAGE – Département Sciences de l'Atmosphère et Génie de l'Environnement, 59000 Lille,
France
[4] Department of Chemistry, University of Massachusetts - Amherst, Amherst, MA 01003, USA
[5] Department of Chemistry, Drexel University, Philadelphia, PA 19104, USA
* now at California Air Resources Board, Sacramento, CA 95812, USA
** now at Cooperative Institute for Research in Environmental Sciences, University of Colorado, Boulder, CO 80309, USA
and Chemical Sciences Division, Earth System Research Laboratory, National Oceanic and Atmospheric Administration,
Boulder, CO 80305, USA
*** now at Momentive Performance Materials, Inc., Tarrytown, NY 10591, United States
*Correspondence*: Philip S. Stevens (pstevens@indiana.edu)
**Abstract.** Reactions of the hydroxyl (OH) and peroxy radicals (HO₂ and RO₂) play a central role in the chemistry of the
atmosphere. In addition to controlling the lifetimes of many trace gases important to issues of global climate change, OH
radical reactions initiate the oxidation of volatile organic compounds (VOCs) which can lead to the production of ozone and
secondary organic aerosols in the atmosphere. Previous measurements of these radicals in forest environments characterized
by high mixing ratios of isoprene and low mixing ratios of nitrogen oxides (NO$_x$) have shown serious discrepancies with
modeled concentrations. These results bring into question our understanding of the atmospheric chemistry of isoprene and
other biogenic VOCs under low NO$_x$ conditions.
During the summer of 2015, OH and HO₂ radical concentrations as well as total OH reactivity were measured using
Laser-Induced Fluorescence - Fluorescence Assay by Gas Expansion (LIF-FAGE) techniques as part of the Indiana Radical,
Reactivity and Ozone Production Intercomparison (IRRONIC). This campaign took place in a forested area near the Indiana
University, Bloomington campus characterized by high mixing ratios of isoprene and low mixing ratios of NO$_x$. Supporting
measurements of photolysis rates, VOCs, NO$_x$, and other species were used to constrain a zero-dimensional box model based
on the Regional Atmospheric Chemistry Mechanism (RACM2) and the Master Chemical Mechanism (MCM). Using an OH
chemical scavenger technique, the study revealed the presence of an interference with the LIF-FAGE measurements of OH
that increased with both ambient concentrations of ozone and temperature. Subtraction of the interference resulted in measured
OH concentrations that were in better agreement with model predictions, although the model still underestimated the measured
concentrations, likely due to an underestimation of the concentration of NO at this site. Measurements of HO₂ radical





concentrations during the campaign included a fraction of isoprene-based peroxy radicals ($HO_2^* = HO_2 + \alpha RO_2$) and were
found to agree with model predictions. On average, the measured reactivity was consistent with that calculated from measured
OH sinks to within 20%, with modeled oxidation products accounting for the missing reactivity, although significant missing
reactivity (approximately 40% of the total measured reactivity) was observed on some days.

## 1 Introduction

The hydroxyl radical (OH) is one of the primary oxidants in the atmosphere (Levy, 1972). The OH radical initiates the
oxidation of volatile organic compounds (VOCs) that leads to the production of hydroperoxy radicals ($HO_2$) and organic
peroxy radicals ($RO_2$). In the presence of nitrogen oxides ($NO_x = NO + NO_2$), reactions of these radicals can lead to the
production of ozone and secondary organic aerosols in the atmosphere, the primary components of photochemical smog.
Because of their short atmospheric lifetimes, measurements of OH and $HO_2$ (together $HO_x$) and total OH reactivity can provide
a robust test of our understanding of this complex chemistry (Heard and Pilling, 2003).

12        Multiple field campaigns have been conducted over the years measuring OH and $HO_2$ radicals in both urban and

forested environments. Measurements of OH in urban areas characterized by high mixing ratios of $NO_x$ and anthropogenic
VOCs have been generally consistent with model predictions (Ren et al., 2003; Shirley et al., 2006; Kanaya et al., 2007a;
Dusanter et al., 2009b; Hofzumahaus et al., 2009; Griffith et al., 2016), while measurements in remote forested environments
characterized by low mixing ratios of $NO_x$ and high mixing ratios of biogenic VOCs have often been greater than model
predictions (Tan et al., 2001; Lelieveld et al., 2008; Whalley et al., 2011; Rohrer et al., 2014).

18        However, recent measurements by Mao et al. (2012) in a northern California forest using a new chemical scavenging

technique that removes ambient OH before air enters the detection cell revealed a significant interference associated with their
Laser-Induced Fluorescence (LIF) measurements of OH. The unknown interference was a factor of 2 to 3 times higher than
ambient OH concentrations (Mao et al., 2012). Similar results were observed in a boreal forest by Novelli et al. (2014), who
observed an interference using a similar chemical scrubbing technique that was a factor of 3 to 4 times higher than ambient
OH concentrations. One possible source of this observed interference may be the decomposition of Criegee intermediates
produced from the ozonolysis of biogenic emissions in the low-pressure detection cells used by LIF instruments, although the
ambient concentration of these intermediates in the atmosphere may be too low to explain all of the observed interference
(Novelli et al., 2017; Rickly and Stevens, 2018). Another proposed source of the interference is the decomposition of ROOOH
molecules inside the FAGE detection cell formed from the reaction of OH with $RO_2$ radicals (Fittschen et al., 2019).
Nevertheless, interferences associated with measurements of OH could explain part of the discrepancies between measured
and modeled OH concentrations in forested environments. Monitoring potential interferences associated with OH
measurements using LIF techniques may be crucial for understanding the discrepancies between measurements and models.

31        In contrast to measurements of OH, the agreement between measured and modeled $HO_2$ concentrations have been

highly variable. In urban environments, measured $HO_2$ concentrations were sometimes found to agree with model predictions



(Shirley et al., 2006; Emmerson et al., 2007; Dusanter et al., 2009b; Michoud et al., 2012; Lu et al., 2013; Ren et al., 2013;
Griffith et al., 2016), while other times the measurements were found to be both lower (George et al., 1999; Konrad et al.,
2003) and higher than model predictions (Martinez et al., 2003; Ren et al., 2003; Emmerson et al., 2005; Kanaya et al., 2007a;
Chen et al., 2010; Sheehy et al., 2010; Czader et al., 2013; Griffith et al., 2016). In forested environments, measured $HO_2$
concentrations were sometimes found to agree with model predictions (Tan, D. et al., 2001; Ren et al., 2005; 2006), but were
often found to be either lower (Carslaw et al., 2001; Kanaya et al., 2007b; Whalley et al., 2011; Kanaya et al., 2012; Mao et
al., 2012; Griffith et al., 2013), or higher than model predictions (Carslaw et al., 2001; Kubistin et al., 2010; Kim et al., 2013;
Hens et al., 2014). Part of this variability may be due to interferences from alkene and aromatic based $RO_2$ radicals converting
to $HO_2$ in systems that detect $HO_2$ through conversion to OH by addition of NO in the sample cell. The degree to which the
$RO_2$ species can interfere with $HO_2$ measurements has been quantified through laboratory experiments (Fuchs et al., 2011;
Whalley et al., 2013; Lew et al., 2018). The extent of $RO_2$ radical contributions during $HO_2$ measurements in previous
campaigns is unclear.
Total OH reactivity measurements can complement $HO_x$ measurements by providing a constraint on the total loss of
OH that can be compared to that calculated from co-located measurements of OH sinks. Several recent studies have identified
discrepancies between measured and calculated OH reactivity in which the measured values are significantly greater than the
calculated values (Di Carlo et al., 2004; Hansen et al., 2014; Nölscher et al., 2016; Zannoni et al., 2016). This difference has
been attributed to OH loss from unmeasured VOCs and their oxidation products. In general, significant missing OH reactivity
has not been observed as often in urban environments as it has in forested areas, bringing into question our understanding of
the chemistry of biogenic emissions and their oxidation products (Dusanter and Stevens, 2017).
This study reports measurements and model simulations of $HO_x$ radical chemistry as well as OH reactivity for a
forested site located in Bloomington, Indiana during the 2015 IRRONIC (Indiana Radical Reactivity and Ozone productioN
InterComparison) field campaign. This work compares the measured $HO_x$ radical concentrations to model predictions
incorporating the Regional Atmospheric Chemistry Mechanism 2 (RACM2), in addition to a version updated to include the
Leuven Isoprene Mechanism (RACM2-LIM1), as well as the Master Chemical Mechanism versions 3.2 and 3.3.1 in order to
test the ability of each model to reproduce the observed radical concentrations and total OH reactivity.
**2 Experimental section**
**2.1 IRRONIC location and supporting measurements**
The IRRONIC campaign site was located within a mixed deciduous forest (sugar maple, sycamore, tulip polar, ash and hickory
trees) at the Indiana University Research and Teaching Preserve (IU-RTP) field lab (39.1908º N, 86.502º W) located
approximately 2.5 km northeast of the center of the Indiana University campus, and 1 km from the IN 45/46 bypass at the
northern perimeter. The goals of the campaign included an informal intercomparison of peroxy radical measurements by two
different techniques (Kundu et al., 2019), an analysis of ozone production sensitivity at this site (Sklaveniti et al., 2018), a





comparison of measured OH radical reactivity with that calculated from measured VOCs, and a comparison of measured OH, $HO_2$, and $RO_2$ radicals with model predictions. The main biogenic emission within this area was isoprene, with an average daytime maximum mixing ratio of approximately 4 ppb during the campaign. This area exhibited low anthropogenic influence from the campus area, with an average daytime maximum mixing ratio of NO of approximately 315 ppt and an average day-time maximum $NO_2$ mixing ratio of approximately 2 ppb. Measurements were conducted on top of two scaffolding platforms adjacent to the field lab, approximately 1.8 m from the ground. Additional information regarding the field site and the IRRONIC campaign can be found in Sklaveniti et al. (2018) and Kundu et al. (2019).

Table 1 summarizes the major instrumentation employed during the campaign. NO was measured every 10 s using a chemiluminescence instrument (Thermo model 42i-TL, detection limit 50 ppt / 2 min). Periodic problems with the sensor's high voltage power supply that required an eventual replacement limited the coverage of the measurements. $NO_2$ was measured every 1 s by a Cavity Attenuated Phase Shift (CAPS) instrument (detection limit 40 ppt / 10 s), and ozone was measured every 10 sec using a 2B Technologies model 202 UV absorbance instrument (detection limit 3 ppb / 10 s). Further details on the calibration and baseline measurements for the NO, $NO_2$, and $O_3$ measurements are described in Kundu et al. (2019). Nonmethane hydrocarbons, including C2-C10 alkanes and alkenes, butadiene, C6-C9 aromatic compounds, isoprene, α-pinene, and β-pinene, were measured using a thermal desorption GC/FID instrument with a 1.5-h time resolution. Oxygenated VOCs (OVOCs), including C2-C10 aldehydes, C2-C6 ketones, and C2-C4 alcohols, were measured by thermal desorption GC/FID-MS with a 1.5-h time resolution. Offline sampling focused on measurements of oxygenated VOCs including formaldehyde and C2-C6 aldehydes, acetone, MEK, glyoxal and methylglyoxal using DNPH cartridges and HPLC-UV analysis. C6-C16 VOCs including α-pinene, β-pinene, limonene, camphene, heptane-hexadecane, methylpentene-pentadecene were measured using Sorbent cartridges and GC-MS analysis. Measurements of J($NO_2$) were made by spectral radiometry courtesy of the University of Houston. HONO was measured using a newly developed Laser Photofragmentation/Laser-Induced Fluorescence instrument (Bottorff et al., 2015; Bottorff et al., in prep).

## 2.2 HO$_x$ radical measurements

The Indiana University LIF-FAGE instrument (IU-FAGE) has been described in detail previously (Dusanter et al., 2009a Griffith et al., 2013; 2016). In the LIF-FAGE technique, OH radicals are detected by laser-induced fluorescence after expansion of ambient air to low pressure. This extends the OH fluorescence lifetime, allowing temporal filtering of the fluorescence from laser scatter (Heard and Pilling, 2003). Ambient air is expanded through a 0.64 mm diameter orifice located at the top of a cylindrical nozzle (5 cm in diameter and 20 cm long), resulting in a flow rate of approximately 3 SLPM through the sampling nozzle. Two scroll pumps (Edwards XDS 35i) connected in parallel maintain a pressure inside the cell of 5.5 Torr.

The laser system used in this study consisted of a Spectra Physics Navigator II YHP40-532Q that produces approximately 8 W of radiation at 532 nm at a repetition rate of 10 kHz which is used to pump a Sirah Credo Dye laser (255 mg/L of Rhodamine 610 and 80 mg/L of Rhodamine 101 in ethanol), resulting in 40 to 100 mW of radiation at 308 nm. After exiting the dye laser, a fraction of the radiation is focused onto the entrance of a 12-m optical fiber to transmit the radiation to



the sampling cell which was placed on top of the 1.8-m platform adjacent to the field lab. In the detection cell, the laser crosses
the expanded air perpendicular to the flow in a White cell configuration with 24 passes. For this campaign, the laser power
inside the sampling cell ranged from 0.5 to 4.4 mW and was monitored using a photodiode at the exit of the White cell.
OH radicals are excited and detected using the $A^2\Sigma^+ \upsilon' = 0 \leftarrow X^2\Pi \upsilon'' = 0$ transition near 308 nm (Stevens et al., 1994).
The net signal is measured by spectral modulation by tuning the wavelength on- and off-resonance in successive modulation
cycles. A reference cell where OH is produced by thermal dissociation of water vapor is used to ensure that the laser is tuned
on and off the OH transition. The OH fluorescence is detected using a microchannel plate photomultiplier tube (MCP-PMT)
detector (Hamamatsu R5946U-50), a preamplifier (Stanford Research System SR445) and a gated photon counter (Stanford
Research Systems SR 400). The MCP-PMT is switched off during the laser pulse through the use of electronic gating allowing
the OH fluorescence to be temporally filtered from laser scattered light. A Teflon injector located approximately 2.5 cm below
the inlet and 17.5 cm above the detection axis allowed for the addition of NO (approximately 2 sccm, $1.4 \times 10^{13}$ cm$^{-3}$, Matheson
Gas, 10% in $N_2$) to convert ambient $HO_2$ to OH through the fast $HO_2 + NO \rightarrow OH + NO_2$ reaction, allowing for indirect
measurements of $HO_2$.
The IU-FAGE instrument is calibrated by producing known quantities of OH and $HO_2$ from the photolysis of water
vapor in air using a mercury penlamp within the calibration source as described previously (Dusanter et al., 2008). For these
calibrations, zero air was sent through a humidifier and delivered at a flow rate of 38-50 L min$^{-1}$ to the calibration source.
Uncertainties associated with the UV water photolysis calibration technique have been described previously (Dusanter et al.,
2008) and are estimated to be 18% (1σ) for both OH and $HO_2$.

### 2.2.1 Measurement of OH interferences

The LIF-FAGE measurements are subject to potential interferences where OH radicals are generated inside the detection cell.
For example, ozone can be photolyzed by the laser and in the presence of water vapor can produce hydroxyl radicals (Davis
et al., 1981a; 1981b) (reactions R1 and R2):
$\qquad O_3 + h\nu \rightarrow O(^1D) + O_2$ $\hspace{8cm}$ (R1)
$\qquad O(^1D) + H_2O \rightarrow 2OH$ $\hspace{8.3cm}$ (R2)
This interference in the IU-FAGE instrument is monitored through laboratory calibrations utilizing various concentrations of
ozone, water vapor, and laser power. To characterize this and any other interference during ambient measurements, a chemical
scrubbing technique is used to remove ambient OH prior to entering the detection cell (Griffith et al., 2016; Rickly and Stevens,
2018). This chemical modulation technique is used to monitor levels of the laser-generated ozone-water interference and any
other processes that may produce OH radicals within the excitation axis.
Hexafluoropropylene ($C_3F_6$, 95.5% in $N_2$, Matheson) is added through a circular injector 1 cm above the nozzle with
a flow rate of approximately 3.5 sccm to remove 95% of externally generated OH (Rickly and Stevens, 2018). During ambient
measurements, chemical addition of $C_3F_6$ is modulated in between ambient OH measurements every 15 minutes for a duration
of 10 minutes. The differences between the measured OH during $C_3F_6$ addition and OH measurements including the



interference represents the net ambient OH concentration in the atmosphere. Taking the measurement of potential interferences
into account results in a limit of detection for OH for this campaign of approximately $7.9 \times 10^5$ cm$^{-3}$ for a 30 min average (S/N
= 1).

## 2.2.2 Contribution of RO$_2$ interferences during HO$_2$ measurements

As discussed above, HO$_2$ radicals are measured indirectly after sampling ambient air at low pressure through chemical
conversion to OH by addition of NO and subsequent detection of OH by LIF:
$\qquad$ HO$_2$ + NO → OH + NO$_2$ $\hspace{6cm}$ (R3)
It was previously believed that the detection of HO$_2$ radicals using this technique was free from interferences from the reaction
of RO$_2$ radicals with NO, as model simulations and measurements suggested that the rate of conversion of RO$_2$ radicals to HO$_2$
by reactions R4 and R5 and subsequent conversion to OH through reaction R3 were negligible. This was due to the slow rate
of reaction R5 under the reduced oxygen concentration in the low pressure LIF-FAGE cell and the short reaction time between
injection of NO and detection of OH (Heard and Pilling, 2003).
$\qquad$ RO$_2$ + NO → RO + NO$_2$ $\hspace{6cm}$ (R4)
$\qquad$ RO + O$_2$ → R'O + HO$_2$ $\hspace{6cm}$ (R5)
For example, RO$_2$ radicals produced from the OH-initiated oxidation of small alkanes were found to produce a negligible yield
of HO$_2$ (Stevens et al., 1994; Kanaya et al., 2001; Tan, et al., 2001; Creasey et al., 2002; Holland et al., 2003). However, recent
laboratory studies have shown that there are interferences associated with measurements of HO$_2$ from the conversion of RO$_2$
radicals derived from the OH-initiated oxidation of alkenes and aromatics to HO$_2$ (and subsequently OH) by reaction with NO.
The high conversion efficiency of alkene-based peroxy radicals to HO$_2$ is due to the ability of the β-hydroxyalkoxy radicals
produced from OH + VOC reactions to rapidly decompose, forming a hydroxyalkyl radical which then reacts rapidly with O$_2$
leading to the production of a carbonyl compound and HO$_2$ (Fuchs et al., 2011; Whalley et al., 2013; Lew et al., 2018). Because
of this interference, measurements of peroxy radicals that are sensitive to this interference are denoted as HO$_2$* ([HO$_2$*] =
[HO$_2$] + α [RO$_2$], 0<α<1). The conversion efficiency depends on the instrumental characteristics and configurations employed
as well as the amount of NO added. The RO$_2$-to-HO$_2$ conversion efficiencies for a number of different peroxy radicals have
been characterized for current and past configurations of the IU-FAGE instrument (Lew et al., 2018). For the configuration of
the IU-FAGE instrument used in this study, the conversion efficiency of isoprene-based peroxy radicals was found to be
approximately 83%, while the conversion efficiency of propane peroxy radicals was found to be approximately 15%. The
precision for the HO$_2$* measurement does not depend on the RO$_2$ interference and results in a limit of detection for HO$_2$*
during this campaign of $7 \times 10^7$ cm$^{-3}$ for a 30 second average (S/N =1).


## 2.3 OH reactivity measurements

The IU Total OH Loss rate Method (TOHLM) instrument is based on the method of Kovacs and Brune (2001) and is described in detail elsewhere (Hansen et al., 2014). Briefly, the instrument is comprised of a flow tube reactor measuring 5 cm in diameter and 75 cm in length.  Ambient air is introduced through an 8 cm diameter perfluoroalkoxy polymer film hose attached to the flow tube at a flow rate of approximately 180 SLPM using a regenerative blower (Spencer VB001) to establish turbulent flow conditions. Previous measurements have demonstrated that different lengths of this inlet tubing do not significantly impact the measured OH reactivity (Hansen et al., 2014). A pitot-static tube (Dwyer Instruments) is positioned just before the exit of the flow tube facing the turbulent core of the flow, approximately 1 cm from the flow tube wall. The pitot-static tube is connected to a 0-1 Torr differential pressure gauge (MKS Instruments) to measure the total flow tube velocity.

OH radicals are produced in a movable injector that houses a mercury pen lamp (UV Pen-Ray) in which the top of the pen lamp was positioned at the end of the injector, just before a spiral Teflon spray nozzle used to promote mixing within the flow tube (McMaster Carr). In addition, a turbulizer is attached to the injector tube 24 cm before the spray nozzle consisting of four 1 cm wide fins to promote turbulent flow conditions as well as to provide support of the injector throughout the flow tube. The injector is inserted along the main axis and is configured for automated movement acquiring continuous measurements in the forward and backward directions. A nitrogen flow of 10 standard liters per minute (SLPM) is bubbled through high-purity water (EMD Chemicals) producing water vapor which is directed through the injector and photolyzed by the penlamp to produce OH with typical concentrations on the order of $10^9$ cm$^{-3}$. This method is known to also produce $HO_2$ radicals, which can lead to a regeneration of OH at NO mixing ratios greater than 1 ppbv (Kovacs and Brune, 2001). However, because the average NO mixing ratio measured over the course of the campaign was below this value, no correction to the measured reactivity was applied (Hansen et al., 2014).

OH radicals were measured using a similar FAGE detection cell described above. Ambient air was expanded through a 1 mm diameter orifice to a total pressure of approximately 6 Torr. OH radicals were excited by a portion of the 308 nm output of the dye laser, with the resulting fluorescence detected by a gated channel photomultiplier tube detector (Excelitas MP 1300) and monitored by a photon counter (Stanford Research SRS 400). A 2 meter long optical fiber was used to transmit the 308-nm laser beam to the OH reactivity detection cell which was located inside the field lab. The laser power was measured at the exit of the detection cell and monitored with a photodiode.

As ambient air entered the flow tube, the automated OH source injector allowed for varying reaction time with the ambient air over a distance of approximately 15 cm for a period of 2.5 minutes. This produced an OH decay over a reaction time of 0-0.15 s from which the OH reactivity was determined. Losses of OH on the walls of the flow tube were measured by flowing high-purity nitrogen (Indiana Oxygen) at 180 SLPM through the flow tube in addition to the OH production through the injector to measure the decay of OH in the absence of any VOCs. Several measurements of this wall loss ($k_b$) resulted in an average value of $10 \pm 2$ s$^{-1}$ (1$\sigma$).



The calculated OH reactivity for a measured compound X ($k_X$), can be determined from the product of the
concentration of X and its second-order rate constant with OH:
$$k_X = k_{OH+X}[X] \tag{1}$$
Summation of this value for each reacting species gives the total OH reactivity ($k_{OH}$):
$$k_{OH} = \sum_i k_{OH+X_i}[X_i] \tag{2}$$
Under pseudo-first order conditions ([OH]<<[X]), the OH concentration within the flow tube can be expressed as a first-order
exponential decay:
$$[OH]_t = [OH]_0 e^{-(k_{OH}+k_b)t} \tag{3}$$
Solving for $k_{OH}$, the OH reactivity, gives:
$$k_{OH} = -\frac{\Delta \ln[OH]}{\Delta t} - k_b \tag{4}$$
Measurements of the change in the concentration of OH over the reaction time produces the measured OH reactivity value.
These measurements can be compared to the calculated total reactivity from measured OH sinks (Eq. 2) to determine whether
the measured total OH reactivity can be accounted for by the measured sinks. The difference between the measured and
calculated total OH reactivity is referred to as the "missing" OH reactivity.
Laboratory measurements of the reactivity of several VOCs with well-known rate constants showed that the OH
reactivity measurements are on average 30% lower than calculated when the measured velocity of the turbulent core is used
to determine the reaction time, likely due to either incomplete mixing of the reactants or a systematic underestimation of the
reaction time (Hansen et al, 2014). As a result, the measured ambient OH reactivity values were scaled by a factor of 1.41.
Measurements performed over a range of OH reactivity values suggest that the IU-TOHLM instrument can measure OH
reactivity up to 45 s$^{-1}$ with a precision (1σ) of 1.2 s$^{-1}$ + 4% of the measured value for a 10 min average (Hansen et al., 2014).
**2.4 Modeling HO$_x$ concentrations and OH reactivity**
Ambient measurements of OH, HO$_2$*, and total OH reactivity were modeled with the Regional Atmospheric Chemistry
Mechanism (RACM2) (Goliff et al., 2013) and the Master Chemical Mechanism version 3.2 (Jenkin et al., 1997; Saunders et
al., 2003). The isoprene oxidation mechanism in RACM2 was updated to include the Leuven Isoprene Mechanism (LIM1)
originally proposed by Peeters, et al. (2009) involving peroxy radical isomerization reactions leading to additional HO$_x$ radical
production (Tan et al., 2017). The addition also includes a revision of the chemistry of first-generation isoprene oxidation
products, including methyl vinyl ketone (MVK), methacrolein (MACR), and isoprene hydroperoxides (ISHP) (Tan et al.,
2017). In addition, the ambient measurements were also modeled with version 3.3.1 of the Master Chemical Mechanism





(MCM). In comparison to MCM 3.2, MCM 3.3.1 includes an updated isoprene oxidation mechanism based on the LIM
mechanism resulting in $HO_x$ recycling from peroxy radical H-shift isomerization reactions (Jenkin et al., 2015).

3        The Framework for 0-D Atmospheric Modeling (F0AM) was used to calculate the radical concentrations and OH

reactivity observed at the IRRONIC site (Wolfe et al., 2016). The model was constrained by the 30 minute average measured
mixing ratios of ozone, $NO_x$, and VOCs and processed through a 5 day spin-up to generate unmeasured secondary oxidation
products. Table S1 summarizes the measured compounds and includes their grouping into the condensed RACM2 model
inputs. Because the VOC measurements occurred every 90 minutes, the measurements were interpolated into 30 min bins
before input to the model. Due to the minimal overlap of the NO measurements with the $HO_x$ measurements, the model was
constrained to the measured diurnal averaged mixing ratio of NO for all days. The measured $J(NO_2)$ was used to scale the
model calculated $J(NO_2)$ and other photolysis rates. The model uncertainty is approximately 30% ($1\sigma$), estimated from
uncertainties associated with the input parameters and the rate constants for each reaction (Griffith et al., 2013; Wolfe et al.,

12 2016).

## 3 Results and discussion

Campaign diurnal average measurements of $J(NO_2)$, temperature, isoprene, $O_3$, $NO_2$, and NO are summarized in Fig. 1. The
maximum average mixing ratio of NO of approximately 315 ppt was observed at approximately 08:00 (EDT), while the
average mixing ratio of $NO_2$ reached a maximum of 2 ppb around 10:00. Average mixing ratios of isoprene ranged from 0.4
to 4.4 ppb, reaching a maximum around 18:00. Anthropogenic VOCs were relatively low at this site, with maximum mixing
ratios of benzene less than 80 ppt. Day-to-day profiles (July 10 to July 25) are illustrated in Fig. 2, showing measurements of
$O_3$, temperature, isoprene, $NO_x$, $HO_2$*, and OH. Unfortunately, instrumental problems limited the NO measurements prior to
19 July.

### 3.1 OH measurements and model comparison

OH concentrations were determined using the chemical modulation technique described above using external $C_3F_6$ addition to
scavenge ambient OH and measure interferences producing OH inside the IU-FAGE detection cell, including laser generated
OH. The measured interferences were subtracted from the total OH signal determined from spectral modulation, resulting in
net ambient OH concentrations (Fig. 2). As can be seen from this figure, the measured interference was a significant fraction
of the total OH signal on many days. On average the measured interference (including laser-generated OH from equations R1
and R2) accounted for approximately 50% of the total signal during the day (08:00-20:00) and as much as 100% of the signal
at night.

29        Figure 3 illustrates the total measured OH radical signal by spectral modulation (black circles), the measured

interference (blue squares), and the expected laser-generated interference from reactions 3 and 4 calculated from laboratory
calibrations (Griffith et al., 2016) (green points) during 14 July and 15 July. On 15 July, the measured interference was similar





to the calculated interference suggesting that the majority of the measured interference was laser-generated. However, on 14
July, the measured interference was much larger than the calculated interference, suggesting that the majority of the measured
interference was due to an unknown source. Subtraction of the calculated laser-generated interference from the measured
interference on all days resulted in a measurement of the unknown interference that increased with both ozone and temperature
during the campaign (Fig. 4).
This result is consistent with the observations from Mao et al. (2012) who found that the interference measured in
their LIF-FAGE instrument using a similar chemical modulation technique increased with ozone and total OH reactivity. The
observed increase in the magnitude of the unknown interference with ozone and temperature suggests that the interference
may be related to the ozonolysis of biogenic VOCs, whose emissions increase with temperature. Previous measurements have
shown that some LIF-FAGE instruments, including the IU-FAGE instrument, are susceptible to an interference under high
concentrations of ozone and biogenic VOCs, perhaps due to the decomposition of Criegee intermediates inside the FAGE
detection cell (Fuchs et al., 2016; Novelli et al., 2017; Rickly and Stevens, 2018). However, estimated concentrations of
Criegee intermediates in similar environments on the order of $5 \times 10^4$ cm$^{-3}$ (Novelli et al., 2017) are too low to explain the
observed interference during the IRRONIC campaign.
The observation of a significant interference during this campaign is in contrast to previous measurements of OH by
the IU-FAGE instrument in a forested environment during the CABINEX 2009 campaign (Griffth et al., 2013). During this
campaign, several tests were conducted where $C_3F_6$ or CO was added to remove ambient OH. These tests did not reveal any
significant interference, and measurements of OH were found to be in good agreement with model predictions (Griffith et al.,
2013). One possible explanation for this discrepancy with the measurements during IRRONIC is the lower levels of ozone and
temperatures observed during CABINEX compared to IRRONIC. Average mixing ratios of ozone during CABINEX were
near 30 ppb and average temperatures were near 20°C during the day, with average mixing ratios of isoprene less than 2 ppb
in the afternoon. These levels of ozone and temperature are lower than that where the interference was observed during
IRRONIC (Fig. 4), suggesting that a similar interference was likely undetectable during CABINEX.
Recent measurements have found that NO$_3$ radicals can lead to an interference in FAGE instruments (Fuchs et al.,
2016), although the mechanism for production of this interference is not known. Such an interference in the IU-FAGE
instrument could explain the observed interference during some nights (Fig. 3), but is unlikely the source of the interference
during the daytime. Another possible source of the interference is the decomposition of ROOOH molecules inside the FAGE
detection cell formed from the reaction of OH with RO$_2$ radicals (Fittschen et al., 2019). However, assuming a rate constant
of $1 \times 10^{-10}$ cm$^3$ s$^{-1}$ for the OH + RO$_2$ reaction, it is unlikely that a significant fraction of RO$_2$ radicals will react to form
ROOOH under the mixing ratios of NO observed at this site, as the estimated lifetime of RO$_2$ radicals with respect to reaction
with NO was an order-of-magnitude shorter than that for reaction with OH. Additional measurements and laboratory tests will
be needed to identify and minimize interferences associated with LIF-FAGE measurements of OH.
The day-to-day measurements of OH after the interference has been subtracted for 10-20 July and 24-25 July are
illustrated in Fig. 5. Measurements on 21-22 July focused on measurements of HO$_2$*, thus OH measurements were not





conducted on those days. This figure also illustrates the day-to-day model results for OH and HO$_2$* from the base RACM2
and the modified RACM2-LIM1 models, as well as the MCM versions 3.2 and 3.3.1, illustrating that, the predicted OH
concentrations are generally lower than the measured concentrations for both the RACM2 and MCM models.

4      Figure 6 (top) shows the average diurnal profile of the OH measurements, both with and without the measured

interference for the days illustrated in Fig. 5. The average ambient diurnal OH radical concentration reached a maximum of
approximately 4-5 × 10$^6$ cm$^{-3}$ after the measured interference was subtracted. If the measured interference was not subtracted
from the total OH signal determined by spectral modulation, the resulting OH radical concentrations would be as high as 9 ×
10$^6$ cm$^{-3}$ (Fig. 6), much greater than the averaged RACM2 and MCM modeled maximum concentrations of approximately 2
× 10$^6$ cm$^{-3}$. The daytime OH radical concentration measurements after the interference has been subtracted are in better
agreement with the model results, but are still approximately a factor of two times larger from 12:00 to midnight and appear
to peak later than the model predictions. Including versions of the LIM1 mechanism for HO$_x$ regeneration in both the RACM2
model (RACM2-LIM1) and the MCM (MCM 3.3.1) results in somewhat higher modeled daytime concentrations of OH
compared to the base RACM2 and MCM 3.2 mechanisms, although the results are still lower than the measured concentrations
(Fig. 6). However, as seen in Fig. 6, if the measured interference was not subtracted, OH radical concentrations would be a
factor of 4-5 times higher than the model predictions.

16     A possible reason for the model underprediction of the measurements is an underestimation of the concentration of

NO in the model. As discussed above, instrumental problems limited the measurements of NO primarily to several days at the
end of the campaign, resulting in approximately 3 days that overlapped with the OH measurements (Fig. 2). Consistent
measurements were only obtained after replacing the instrument's detector. In order to model the remaining days of the
campaign, the model was constrained to the diurnal average of the NO measurements from the latter half of the campaign.
However, it is possible that the actual mixing ratio of NO during the early days of the campaign was higher than the average
value measured during the end of the campaign, given that the measured NO$_2$ concentrations during the early part of the
campaign were approximately a factor of 2 greater than that measured during the latter part of the campaign (20-24 July) (Fig.
2). For the days at the end of the campaign where there was significant overlap between the measurements of OH and NO, the
model results are in better agreement during these days (20 and 24 July) (Fig. 5). The diurnal average model results are in
better agreement with the measurements when mixing ratios of NO were unconstrained while constraining mixing ratios of
NO$_2$ and O$_3$. As shown in Fig. 6, unconstraining the concentration of NO in the MCM 3.3.1 model increases the predicted OH
concentrations by approximately a factor of 3 during the daytime with model predicted mixing ratios of NO approximately a
factor of 2 greater than the constrained values during the day. Although the model still underestimates the measurements of
OH in the afternoon, it is clear that without taking the observed OH interference into account, the measured OH concentrations
would have been a factor of 5 greater than predicted by the model mechanisms, similar to previous measurements under
comparable mixing ratios of isoprene and NO$_x$ (Rhorer et al., 2014).





## 3.2 HO₂* measurements and model comparison

The day-to-day measurements of $HO_2^*$ are illustrated in Fig. 5 with the RACM2, RACM2-LIM1, MCM 3.2 and MCM 3.3.1 model results. The contribution of modeled $RO_2$ radicals to the modeled $HO_2^*$ is based on laboratory calibrations of the $RO_2$–to–$HO_2$ conversion efficiencies for the sampling conditions used in this study (Lew et al., 2018) and are incorporated into both versions of the RACM2, and MCM peroxy radical categories. Under the instrumental conditions during the campaign, the conversion efficiency of isoprene-based peroxy radicals to $HO_2$ was determined to be approximately $83 \pm 7\%$, while the conversion efficiency of methyl peroxy radicals was estimated to be approximately 5% (Lew et al., 2018). These two peroxy radicals accounted for the majority of $RO_2$ radicals predicted by the models (see below). The maximum measured $HO_2^*$ concentration each day during the campaign was generally between approximately $2 \times 10^8$ and $2 \times 10^9$ molecules cm⁻³ (Figs. 2 and 5), with an average daily maximum value of approximately $1 \times 10^9$ cm⁻³ (Fig. 6). The RACM2-LIM1 and MCM 3.3.1 modeled diurnal averaged $HO_2^*$ reached a maximum of approximately $1.3 \times 10^9$ cm⁻³ and $9.5 \times 10^8$ cm⁻³, respectively, compared to a value of $1.2 \times 10^9$ cm⁻³ for the RACM2 modeled $HO_2^*$ and $9.1 \times 10^8$ molecules cm⁻³ for the MCM 3.2 modeled $HO_2^*$ (Fig. 6).

The predicted $HO_2^*$concentrations by the base RACM2 model are in good agreement with the measured concentrations, overpredicting the measurements by approximately 20% on average, although the model agrees with the measurements to within the combined uncertainty of the model and the measurements. Including the LIM1 mechanism in the RACM2 mechanism increases the modeled $HO_2^*$ by approximately 15% due to the modeled increase in $HO_x$ radical production from the isomerization of isoprene-based peroxy radicals. The MCM-based model results are also in good agreement with the measured $HO_2^*$ although they tend to underpredict the measured concentrations by approximately 20% on average in the afternoon (Fig. 5 and 6). The MCM 3.3.1 mechanism results in predicted $HO_2^*$ concentrations that are approximately 5% greater than that predicted by MCM 3.2 in the afternoon when NO concentrations are low due to the inclusion of $HO_x$ production from the isomerization of isoprene-based peroxy radicals. These results are also consistent with a possible under-estimation of the actual concentrations of NO at the site as discussed above. Unconstraining the mixing ratio of NO in the MCM 3.3.1 model increases the averaged modeled $HO_2^*$ concentrations to values similar to that predicted by the RACM2 model, but still within approximately 20% of the measured concentrations and in better agreement with the measurements in the late afternoon (Fig. 6). These results are in contrast to that observed during the CABINEX campaign, where a RACM-based model overpredicted the measured $HO_2^*$ by as much as a factor of 2 (Griffith et al., 2013), likely related to the higher concentrations of NO observed during IRRONIC compared to CABINEX increasing the importance of the $HO_2 + NO$ and $RO_2 + NO$ reactions in determining the fate of these radicals.

The MCM 3.2 and MCM 3.3.1 diurnal average modeled $HO_2^*$ concentrations and the model contribution of peroxy radicals to $HO_2^*$ are shown in Fig. 7 (left panels). The diurnal profile of the $HO_2^*$ radical concentration predicted by the MCM models includes contributions primarily from isoprene peroxy radicals and $HO_2$ radicals, with smaller contributions from methyl peroxy and acetyl peroxy radicals (Fig. 7). The RACM2 models produced similar results, with $HO_2$ and isoprene peroxy





radicals contributing to the majority of the modeled $HO_2^*$ concentrations (Fig S1). The total modeled $RO_x$ ($RO_2$ + $HO_2$)
concentrations by the different mechanisms are also shown in Fig. 7 (right panels). The MCM 3.2 model predicted that the
diurnal average total $RO_x$ concentration consisted primarily of $HO_2$ (52%), isoprene peroxy radicals (20%), methyl peroxy
($CH_3O_2$, 22%), and acetyl peroxy ($CH_3CO_3$, 5%), with daytime (08:00 – 20:00) contributions of 48%, 26%, 19%, and 5% for
$HO_2$, isoprene peroxy, $CH_3O_2$, and $CH_3CO_3$, respectively. The MCM 3.3.1 model predicted that $HO_2$ (53%), isoprene peroxy
(16%), methyl peroxy (23%), acetyl peroxy (5%) were the major contributors to the modeled diurnal average total $RO_x$
concentration, with daytime contributions of 50%, 22%, 21%, and 6% (Fig. 7). Similar results were obtained from the RACM2
models (Fig S1). As discussed above, the configuration of the IU-FAGE instrument used in this study converted approximately
83% of isoprene peroxy radicals to $HO_2$ upon addition of NO and minimally converts methyl peroxy radicals to $HO_2$ (<5%)
(Lew et al., 2018). Thus, the majority of the contributing species to the measured $HO_2^*$ are $HO_2$ and isoprene peroxy radicals
which together account for approximately 70% of the total peroxy radical concentration predicted by these models.
Measurements of the total $HO_2$ + $RO_2$ radical concentrations using an Ethane – Nitric Oxide Chemical Amplifier (ECHAMP)
were found to be in good agreement with the $HO_2^*$ measurements reported here and are summarized in Kundu et al. (2019).
**3.3 Total OH reactivity measurements and model comparison**

15         The measured total OH reactivity and that calculated from measured OH sinks using both the RACM and MCM

mechanisms are shown in Fig. 8, where the measured OH reactivity is averaged into 2 hour bins. As illustrated in this figure,
the calculated OH reactivity was in relatively good agreement with the measured OH reactivity on some days and nights,
specifically 15-16 July, with missing reactivity observed later in the campaign. Overall, the averaged measured OH reactivity
varied between the instrumental limit of detection of 1 $s^{-1}$ to a maximum of approximately 31$s^{-1}$ with an overall diurnal average
value of approximately 13 $s^{-1}$.

21         The campaign diurnal averaged measured OH reactivity is shown in Fig. 9 along with the calculated total OH

reactivity from the measured OH sinks. On average, the calculated reactivity is in good agreement with the measurements. As
expected for this deciduous forest environment, isoprene was the dominant contributor making up 37% of the diurnally
averaged total reactivity, followed by OVOCs (28%), inorganics (10%), alkanes and alkenes (5%), anthropogenic non-methane
hydrocarbons (NMHC) (1%), and monoterpenes (<1%) with missing reactivity accounting for the remaining 18% (Fig. S2).
During the daytime (08:00 and 20:00) the contributions are similar, with isoprene being the largest contributor at 47% followed
by OVOCs (24%), inorganics (8%), alkanes and alkenes (4%), anthropogenic NMHC (1%), and monoterpenes (<1%) with
missing reactivity accounting for the remaining 14%. During the nighttime, (20:00 to 08:00), OVOCs were the dominant
contributor to the modeled OH reactivity at 32% followed by isoprene (24%), inorganics (11%), alkanes and alkenes (6%),
anthropogenic NMHC (2%), and monoterpenes (<1%) with missing reactivity of 24% (Fig. S2).

31         The campaign diurnal average (Fig. 9) shows a correlation with temperature, with the maximum average OH

reactivity of approximately 20 $s^{-1}$ occurring around 13:30. The calculated reactivity was consistent with the measured reactivity
for temperatures less than 294 K, while the observed reactivity is greater than that calculated from the measured sinks for



higher temperatures, although at temperatures above 302 K the measured reactivity appears to be less than calculated (Fig S3).
These results are similar to that reported by Hansen et al. (2014) and Di Carlo et al. (2004) in which the measured missing
reactivity appeared to increase with temperature.
Figure 9 also shows the campaign average OH reactivity including the reactivity of unmeasured oxidation products
predicted by the MCM 3.3.1 model. On average, including the contribution of unmeasured oxidation products can account for
the majority of the missing reactivity. While the model tends to overpredict the average measured reactivity in the afternoon
and evening, the model results agree to within the combined uncertainty of the model and the precision of the measurement
(Hansen et al., 2014). Similar results were obtained by the RACM2 models, although the predicted reactivity of unmeasured
oxidation products by the RACM2 models are approximately a factor of two smaller than that predicted by the MCM models
(Fig. S4). These results suggest that the models are generally able to reproduce the measured OH reactivity at this site, and
that the missing reactivity observed during IRRONIC may be due to unmeasured oxidation products, with isoprene nitrates
and isoprene epoxides within the RACM2 and MCM mechanisms being the primary contributors to the missing reactivity.
While the campaign averaged OH reactivity measurements appear to be in reasonable agreement with the calculated
reactivity based on measured compounds, there were several days that displayed large missing reactivity similar to that
observed by Hansen et al. (2014). The MCM 3.3.1 model results for a day with the largest missing reactivity (17 July) is shown
in Fig. 10, indicating that the modeled reactivity including unmeasured oxidation products cannot explain the observed
reactivity on this day. The reason for this discrepancy is unclear, but may indicate the presence of additional unmeasured
emissions or oxidation products not accounted for by the model.
**3.4 Radical budgets**
The analysis of the rates of radical initiation, propagation, and termination can provide insight to the importance of
individual radical sources and sinks. For the IRRONIC campaign, the OH radical budget is illustrated in Fig. 11, where OH
radical production reactions are represented in shades of blue and loss reactions are represented in shades of red. Daytime
production includes reactions with both initiation and propagation that produces OH radicals (positive rates), while daytime
OH loss reactions are represented by propagation and termination reactions that remove OH (negative rates). For simplicity
only the RACM2 and RACM2-LIM1 radical budgets are shown.
The maximum rates for the OH radical budget of approximately $2.8 \times 10^7$ cm$^{-3}$ s$^{-1}$ from the RACM2-LIM1 model
were higher than the maximum value of $2.2 \times 10^7$ cm$^{-3}$ s$^{-1}$ in RACM2. The addition of the LIM1 mechanism increases the OH
radical production rate mostly from photolysis of hydroxyperoxy aldehydes (HPALD) produced from the isomerization of
isoprene-based peroxy radicals and their subsequent chemistry (Peeters et al., 2014; Tan et al., 2017). In the RACM2-LIM1
model, the daytime OH radical production is dominated by the HO$_2$ + NO reaction from 10:00 to 14:00 (57%) and drops to
28% from 14:00 to 18:00. Ozone photolysis and the LIM1 mechanism contribute up to 24% and 31% of the total OH radical
production from 14:00 to 18:00, with ozonolysis (VOC+O$_3$) and photolysis of HONO, H$_2$O$_2$, methacrolein (MACR), and
organic peroxides (OP1, OP2) contributing to 13% and 4% of the total OH radical production in the afternoon (Fig. 11). A



majority of the OH radical loss is due to OH reactions with VOCs (66-72%) and OVOCs (22-19%) during the morning and
afternoon. As described above, the measured total OH reactivity was in reasonable agreement with the modeled OH reactivity;
therefore, it is likely that the total OH loss is well represented in the model.

4        The total radical ($RO_x$) budget from the RACM2 mechanisms of OH, $HO_2$, and $RO_2$ radicals is illustrated in Fig. 12.

Overall, total radical initiation in the RACM2-LIM1 mechanism was larger, with a maximum value of approximately 2.6 ×
$10^7$ cm$^{-3}$ s$^{-1}$ compared to RACM2 maximum value of approximately $1.7 \times 10^7$ cm$^{-3}$ s$^{-1}$. The increase in total radical initiation
in the RACM2-LIM1 model is due to both the added radical initiation from the photolysis of HPALDs as well as increased
radical initiation from other aldehydes produced in the LIM1 mechanism. Overall, radical initiation from the photolysis of
HPALDs and the subsequent chemistry from the LIM1 mechanism contributed 8-11% of total radical initiation during the day,
while photolysis of formaldehyde and other aldehydes contributed to approximately 42% of total radical initiation, with ozone
photolysis contributing to 34-37% of radical initiation in the mornings and afternoon (Fig. 12). In contrast, ozone photolysis
contributes to approximately 50% of radical initiation in the RACM2 mechanism compared to formaldehyde and other
aldehydes contributing 31-34% (Fig. 12). Radical termination for both mechanisms is dominated by peroxy radical self-
reactions, such as the $HO_2 + HO_2$ reaction, as well as the reaction of $HO_2$ with isoprene-based peroxy radicals (ISOP) and
other peroxy radicals ($RO_2$). These reactions account for approximately 90-95% of radical termination due to the low levels
of $NO_x$ used in the models, with reaction of OH +$NO_2$ and other $NO_x$ radical reactions accounting for approximately 5-10%
of radical termination in these models (Fig. 12). As discussed above, it is possible that the NO concentration used to constrain
the model may be lower than the actual concentration. As a result, the modeled contribution of $NO_x$ reactions to radical
termination may represent a lower limit to the actual contribution.

20       The partitioning of the total radical budget production for IRRONIC is similar to the modeled budget observed during

PROPHET 2008 and CABINEX 2009 (Griffith et al., 2013). The updated RACM model used during these campaigns predicted
that radical termination was dominated by $HO_2 + RO_2$ reactions (including the HO2 + ISOP reaction), contributing to
approximately 80% of total radical termination, similar to the 70-78% for the $HO_2$+ISOP and $HO_2$+$RO_2$ reactions predicted
here by the RACM2 model. The photolysis of ozone accounted for approximately 20-30% of total radical initiation during
these campaigns based on an updated version of the RACM model (Griffith et al., 2013) compared to approximately 50%
predicted by the RACM2 mechanism during IRRONIC due to higher concentrations observed during this campaign.
Ozonolysis reactions contributed to approximately 20-30% of total radical initiation during PROPHET and CABINEX
compared to 10-14% during IRRONIC. Photolysis of aldehydes, including HCHO, contributed to approximately 30% of the
total rate of radical initiation during IRRONIC compared to 23% and 5% during PROPHET 2008 and CABINEX 2009,
respectively, with the low contribution during CABINEX primarily due to the lower mixing ratios of HCHO observed during
this campaign (Griffith et al., 2013). In contrast, photolysis of HONO was a significant radical source during PROPHET and
CABINEX, contributing 14-17% of radical initiation compared to approximately 5% of total radical production during
IRRONIC due to the lower mixing ratios of HONO observed during IRRONIC. On average, mixing ratios of HONO during
IRRONIC were approximately 40 ppt at night decreasing to approximately 10 ppt during the day (Fig. S5) compared to daytime





mixing ratios between 50 and 75 ppt during PROPHET and CABINEX (Griffith et al., 2013). The reason for the difference in
the measured HONO values between these two sites is unclear, but may be related to increased production from photolysis of
nitric acid on the forest canopy surfaces at the PROPHET site (Zhou et al., 2011).
**4 Summary**
Measurements of OH radical concentrations using the IU-FAGE instrument during the IRRONIC campaign revealed
a significant unknown interference that appeared to correlate with both temperature and ozone. The average measured OH
radical concentration after the interference was subtracted reached an average daytime maximum of approximately $4\text{-}5 \times 10^6$
$cm^{-3}$. This is in contrast to the measurements including the interference which reached an average daytime maximum of
approximately $9 \times 10^6$ $cm^{-3}$. Similar OH concentrations were observed at this site in 2017 during an informal intercomparison
between the IU-FAGE instrument and the University of Colorado Chemical Ionization Mass Spectrometry (CIMS) instrument
(Rosales et al., 2018; Reidy et al., 2018).
After subtracting the interference, the OH measurements were in better agreement with model simulations utilizing
the Regional Atmospheric Chemical Mechanism 2 (RACM2) with an updated Leuven Isoprene Mechanism (LIM1) as well as
the Master Chemical Mechanism versions 3.2 and 3.3.1. Both the RACM2-LIM1 and MCM 3.3.1 mechanisms add radical
recycling reactions for isoprene oxidation that increase the modeled OH and peroxy radical concentrations. The addition of
radical recycling by isoprene still resulted in model predictions of OH that were approximately a factor of two lower than the
measured concentrations. One possible explanation for the discrepancy is an underestimation of the mixing ratio of NO during
the campaign, as instrumental difficulties prevented measurements of NO except at the end of the campaign. Unconstraining
the mixing ratios of NO in the model while constraining $NO_2$ and $O_3$ to their measured values leads to an increase in the
modeled mixing ratios of NO resulting in an increase in the average modeled OH concentration by approximately a factor of
2-3, improving the agreement with the measured OH concentrations. These higher values of $NO_x$ are comparable to that
observed at this site in 2017 when measured OH concentrations were similar to that observed here (Rosales et al., 2018; Reidy
et al., 2018). However, it is clear that if the measured interference was not taken into account, the apparent OH concentrations
would have been a factor of 5 greater than predicted by the model mechanisms, comparable to previous measurements under
low $NO_x$ and high isoprene conditions (Rhorer et al., 2014). These results are similar to that reported by Mao et al. (2012) who
found good agreement between their OH measurements and model predictions when measured interferences are taken into
account. However, because of differences in instrument design (geometry, cell pressure, flow, etc.) these interference
measurements may not apply to other LIF-FAGE instruments. However, future OH measurements using the LIF-FAGE
technique should include methods to quantify potential instrumental artifacts.
Measurements of total OH reactivity were in reasonable agreement with that calculated from measured OH sinks,
with isoprene contributing approximately 37% and OVOCs 28% of the diurnally averaged measured reactivity, with 18% of
the measured reactivity missing. However, on average the missing reactivity fraction can be explained by unmeasured





oxidation products, specifically from isoprene nitrates and isoprene epoxides within the RACM2 and MCM mechanisms. This
indicates that these mechanisms are accurately representing the total OH loss at this site.
Measurements of $HO_2$ radicals by the IU-FAGE instrument using chemical conversion to OH by addition of NO has
been shown to be sensitive to alkene-based peroxy radicals (Lew et al., 2018). As a result, the measurements represent a sum
of $HO_2$ and a fraction of $RO_2$ radicals in the atmosphere ($HO_2*$). During the IRRONIC campaign, the measured $HO_2*$
concentration primarily reflected the sum of $HO_2$ and isoprene-based peroxy radicals, which contributed to approximately 70%
of the total modeled peroxy radicals. The average daytime ambient $HO_2*$ measurements reached maximum concentrations of
approximately $1 \times 10^9$ $cm^{-3}$. Both MCM models predicted $HO_2*$ concentrations that were in good agreement with the
measurements, while the RACM mechanisms resulting in predicted concentrations that were approximately 20-35% greater
than the measurements but within the combined uncertainty of both the model and the measurement. These results are also
consistent with an underestimation of the NO concentrations in the model, as increasing the modeled NO resulted in modeled
$HO_2*$ concentrations that were still in good agreement with the measurements. These results are in contrast to some previous
measurements in forest environments where model predictions were found to be significantly greater than measured $HO_2*$
concentrations (Griffith et al., 2013), perhaps as a result of the lower mixing ratios of NO observed at these sites. Additional
measurements are needed in order to resolve this discrepancy, which may be related to a gap in our understanding of peroxy
radical chemistry under low NO conditions.

**Data availability.** Data are available upon request from the corresponding author (pstevens@indiana.edu).

**Competing interests.** The authors declare that they have no conflicts of interest.

**Author contributions.** PS, SD and EW designed the research project. ML, PR, BB, and PS were responsible for the LIF-FAGE OH, $HO_2*$, OH reactivity, and HONO measurements. SK and EW were responsible for the supporting measurements of NO, $NO_2$, and $O_3$. SD, SS, TL, and NL were responsible for the measurements of VOCs and OVOCs. ML, PR, and PS conducted the analysis and photochemical modelling and wrote the paper with feedback from all co-authors. ML and PR contributed equally to the paper.

**Acknowledgements.** This study was supported by the National Science Foundation, grant AGS-1440834 to Indiana University, AGS-1443842 to the University of Massachusetts, and AGS-1719918 to Drexel University. This work was also supported by grants from the Regional Council Nord–Pas-de-Calais through the MESFOZAT project, the French National Research Agency (ANR–11–LABX–0005–01) and the European Regional Development Fund (ERDF) through the CaPPA (Chemical and Physical Properties of the Atmosphere) project, and the Région Hauts-de-France, the Ministère de l'Enseignement Supérieur et de la Recherche and ERDF through the CLIMIBIO project. We would like to thank J. Flynn



(University of Houston) for the spectroradiometer used to obtain the J(NO₂) measurements, and E. Reidy (Indiana University) for conducting some additional modeling.

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





**Table 1: Measurements conducted during the IRRONIC field campaign.**

| Measurement | Instrument | Technique | LOD | Reference |
|---|---|---|---|---|
| OH | LIF-FAGE | Laser-induced fluorescence – fluorescence assay by gas expansion | $8\times10^5$ cm$^{-3}$ / 30 min | Dusanter et al., 2009a; Lew et al., 2018 |
| HO$_2$* | | | $7\times10^7$ cm$^{-3}$ / 20 s | |
| NO | Thermo 42i-TL | Chemiluminescence | 50 ppt / 2 min | |
| NO$_2$ | Aerodyne CAPS | Cavity attenuated phase shift spectroscopy | 40 ppt / 10 s | |
| Ozone | 2B Technologies Model 202 | UV absorbance | 3 ppb / 10 s | |
| OH reactivity | LIF-TOHLM | Total OH Loss Measurement | 1 s$^{-1}$ (10 min) | Hansen et al., 2013 |
| HONO | LP LIF-FAGE | Laser-photofragmentation laser-induced fluorescence | 20 ppt (30 min) | Bottorff et al., in prep |
| NMHCs | Online GC/FID | Gas chromatography with flame ionization detection | 10-100 ppt (1.5 hr) | Badol et al., 2004 |
| OVOCs | Online GC/FID-MS | Gas chromatography with mass spectrometer and FID | 5-100 ppt (1.5 hr) | Roukos et al. (2009) |
| | Off-line Sorbent GC-MS | Sorbent cartridges analyzed by GC-MS | | Detournay et al. (2011); Ait-Helal et al. (2014) |
| | Off-line DNPH HPLC-UV | Dinitrophenylhydrazine cartridges analyzed by high-performance liquid chromatography with UV detection | | |
| J(NO$_2$) | | Spectral Radiometry | $0.3 \times 10^{-4}$ s$^{-1}$ | Shetter and Muller (1999) |





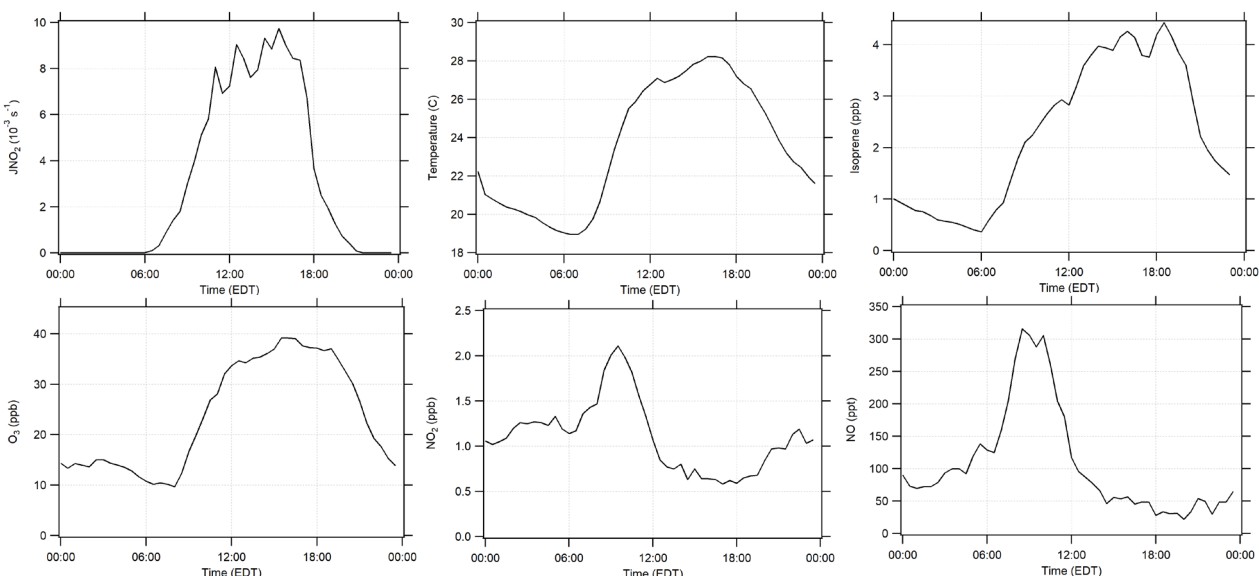

**Figure 1.** Diurnal campaign average profiles of J(NO₂), temperature, isoprene, O₃, NO₂, and NO.



**Figure 2.** Time series of OH and HO₂* from July 10 to July 25 with model calculated J(O¹D) scaled to the measured J(NO₂), and measured ozone, temperature, isoprene, and NOₓ. OH measurements with interference (± 1σ) represented by the green line and measurements without interference (± 1σ) represented by the black line. For clarity, OH data shown are 2 hour averages. HO₂* data are 30 s averages every 30 minutes.





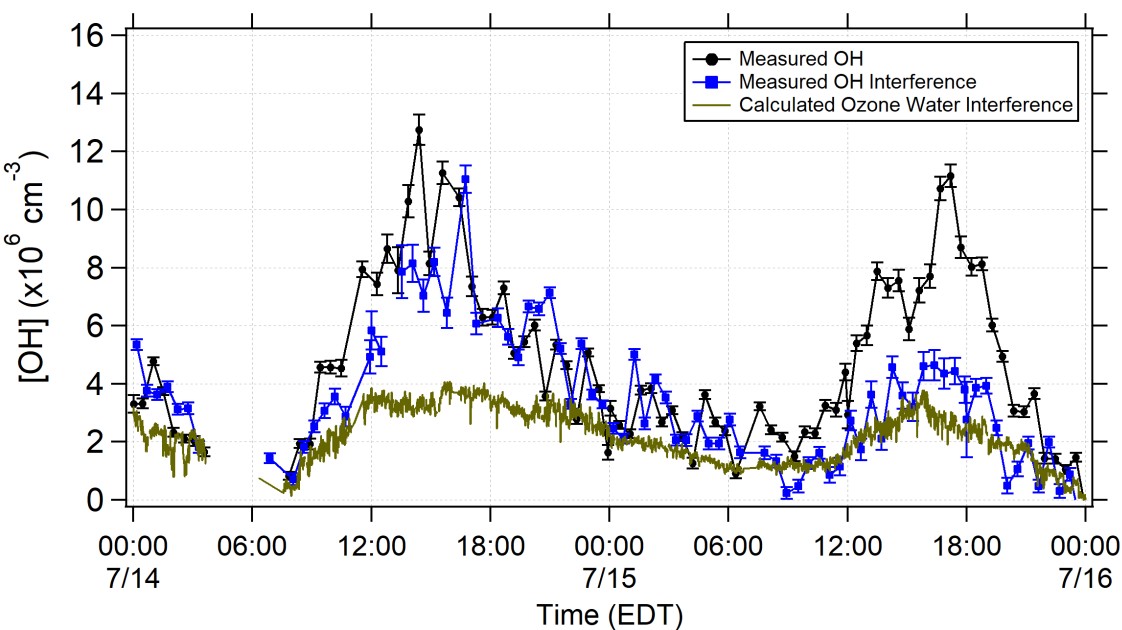

**Figure 3.** Averaged measured total OH signal using spectral modulation (black), and the measured interference using chemical modulation (blue) during July 14 and July 15. The calculated laser-generated interference from ozone photolysis for these days (reactions 1 and 2, green points) is also shown.

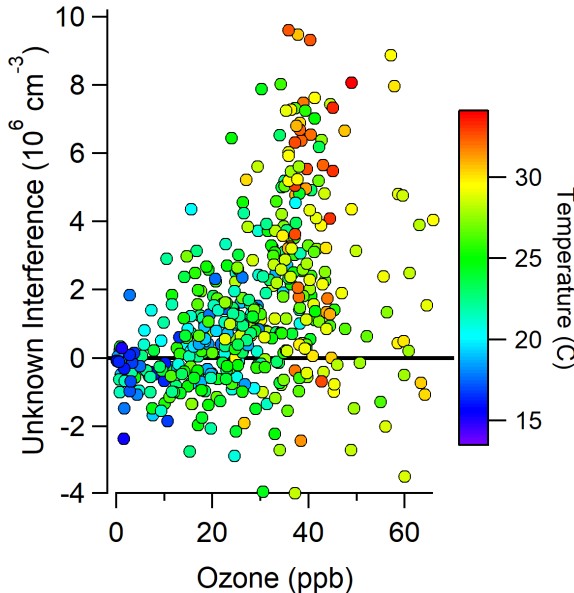

**Figure 4.** Measurements of the unknown interference as a function of ozone and temperature during the campaign.

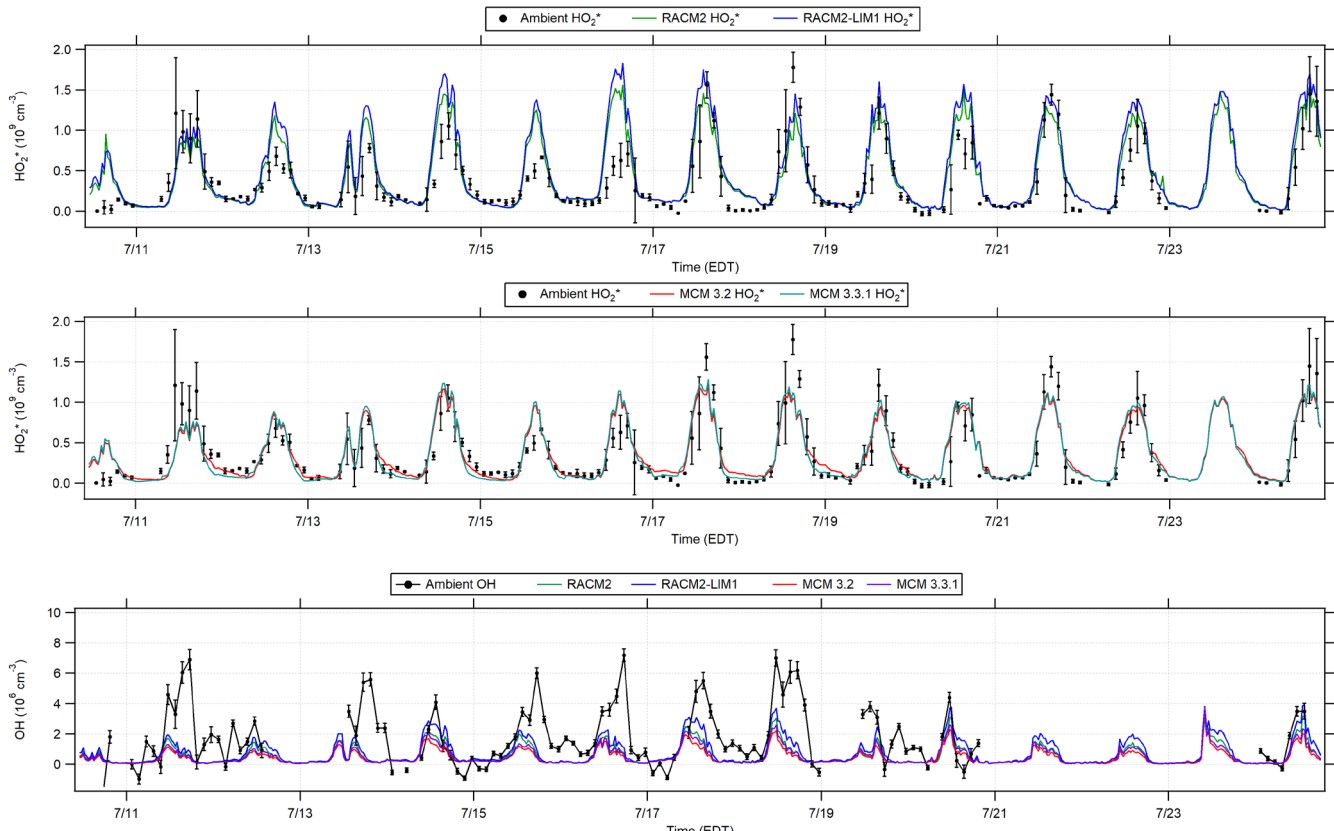

**Figure 5.** Average measurements of OH (bottom) and HO₂* from July 10 to July 25 during the IRRONIC campaign in comparison to modeled results for RACM2 and RACM2-LIM1 models (top) and the MCM 3.2 and MCM 3.3.1 models (middle). The error bars represent the precision of the measurements (1σ).





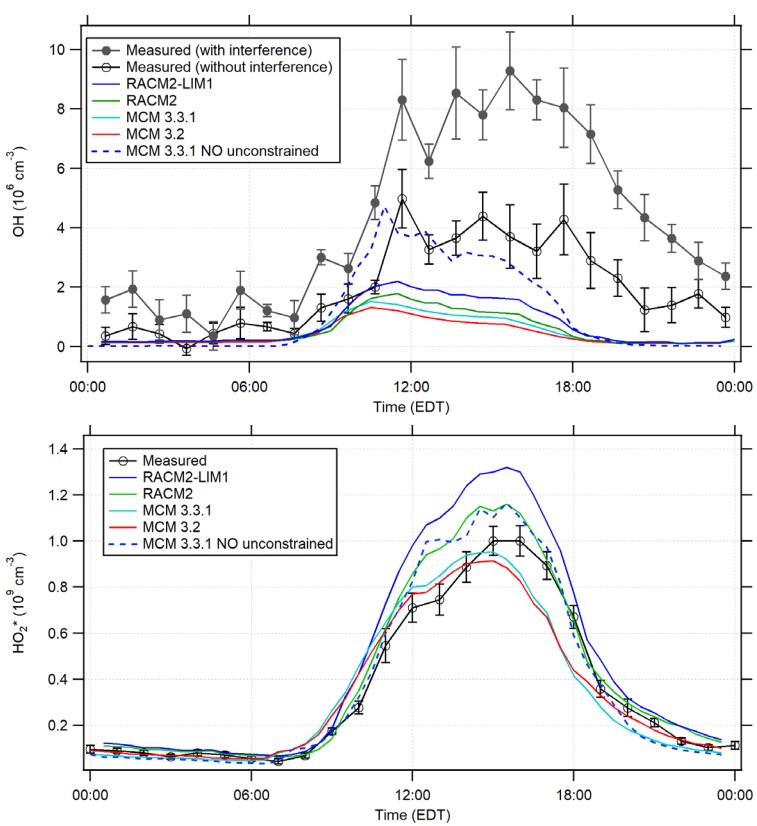

**Figure 6.** Diurnal profiles of OH (top) and HO₂* (bottom) with the RACM2, RACM2-LIM1, MCM 3.2, and MCM 3.3.1 model results. The open circles represent the 1 hour mean ± 1σ standard error of OH and HO₂* measurements. The filled circles represent the 1 hour mean ± 1σ standard error of the OH measurements with the interference. The dashed line represent the MCM 3.3.1 model results with NO concentrations unconstrained (see text).





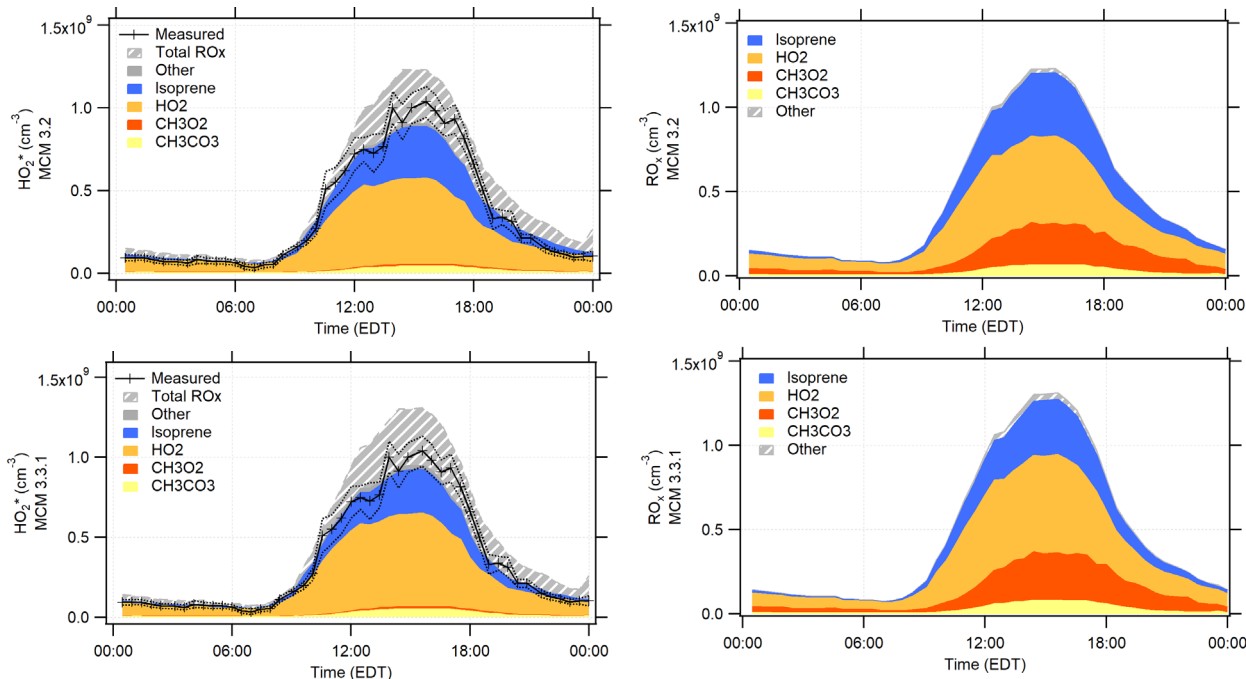

**Figure 7.** The MCM 3.2, and MCM 3.3.1 diurnal average modeled peroxy radical concentration and composition. Left panels show the modeled contribution to the measured $HO_2*$ concentrations. The measured 30-min mean $HO_2*$ concentrations are shown by the black line with ± 1σ standard error shown by the dotted lines. Right panels show the total $RO_x$ ($RO_2$ +$HO_2$) composition predicted by each model.





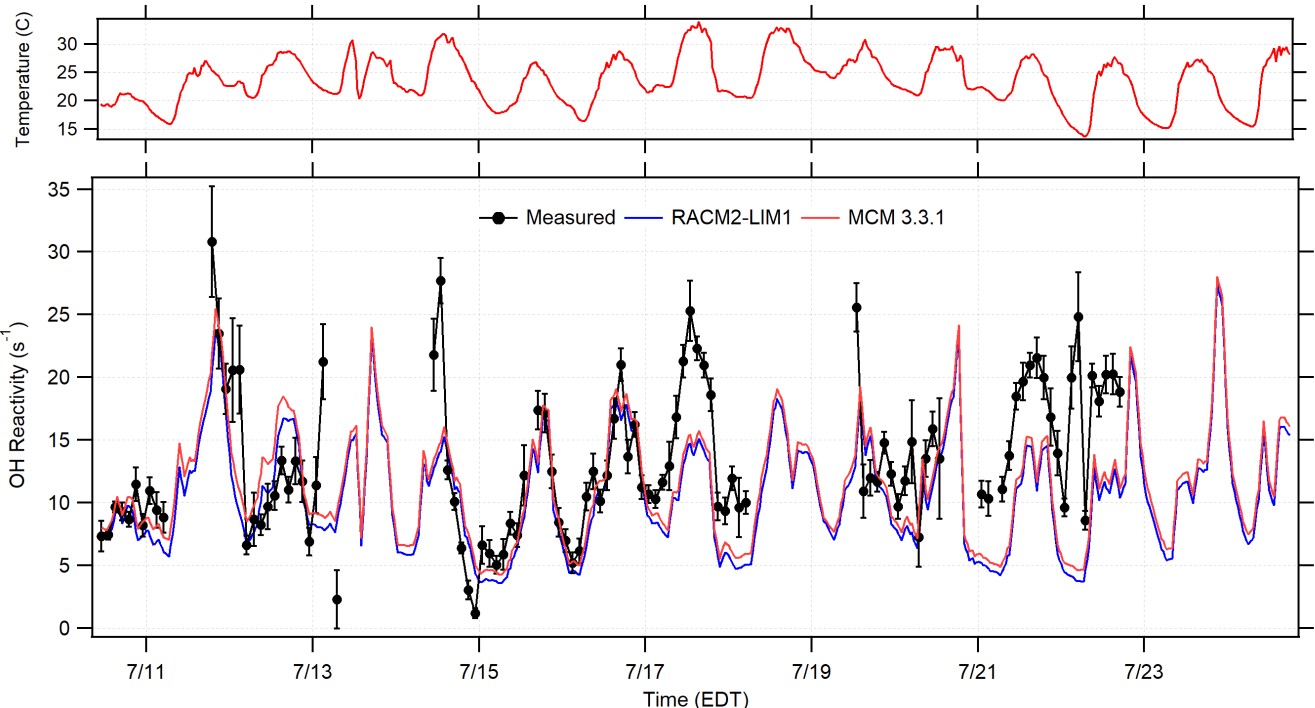

**Figure 8.** Time series of the 2 hour averaged OH reactivity measurements (black circles) in comparison to the RACM2-LIM1 and MCM 3.3.1 calculated OH reactivity based on measured OH sinks along with ambient temperature (top). Error bars represent the standard error of the average measurement.





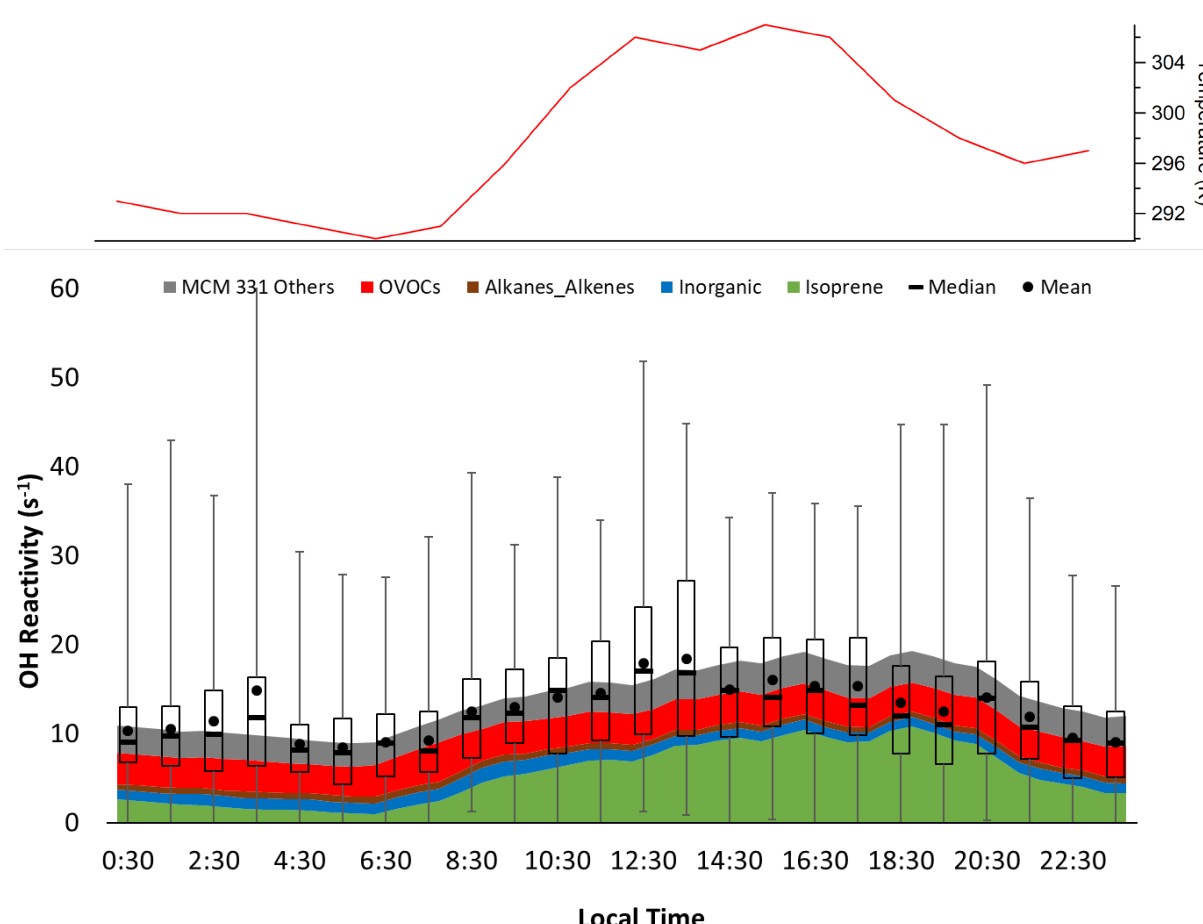

**Figure 9.** Diurnal temperature (top) and box and whiskers plot of observed total OH reactivity showing the mean and median values for each hour, with the mean calculated values from the measured OH sinks as well as the unmeasured oxidation products from the MCM 3.3.1 model results (Others). Error bars show the range of individual 5-min measurements and bars show Q1 and Q3 for the measured OH reactivity.



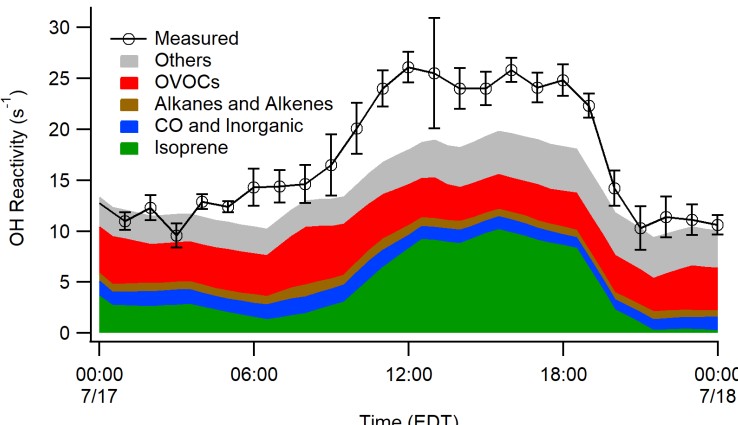

**Figure 10.** Median diurnally averaged OH reactivity from July 17 in comparison to modeled reactivity from the MCM 3.3.1 mechanism.



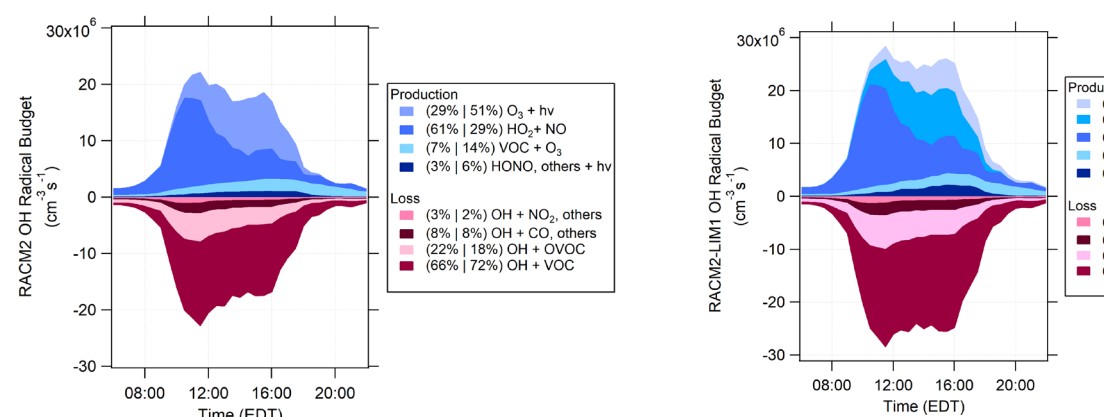

**Figure 11.** RACM2 (left) and RACM2-LIM1 (right) OH radical budgets where the shades of blue represent production reactions and the shades of red represent loss rates. The percent contribution of each reaction to total production/loss are divided into two periods (10:00 to 14:00 and 14:00 to 18:00).

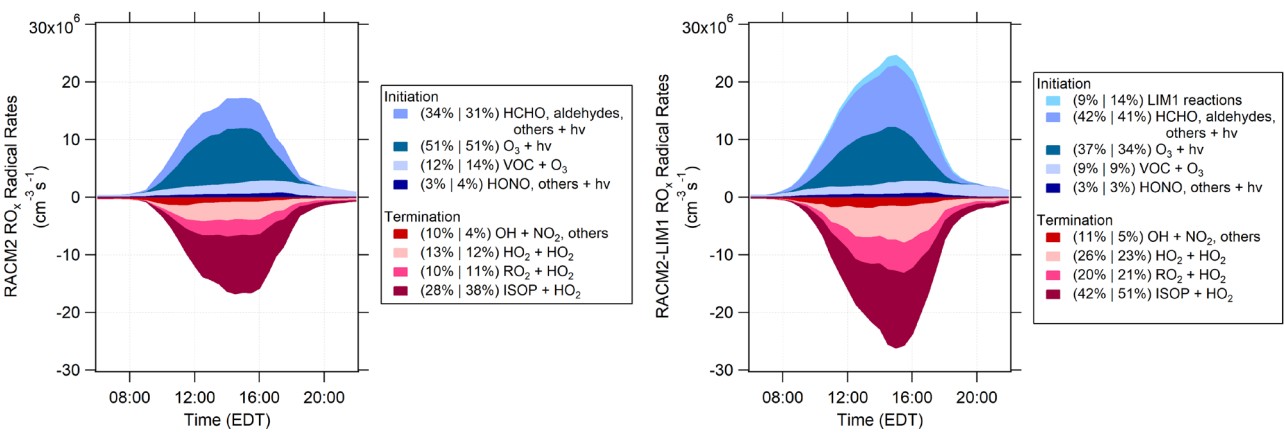

**Figure 12.** RACM2 (left) and RACM2-LIM1 (right) total $RO_x$ radical budgets where the shades of blue represent initiation rates and the shades of red represent termination rates. The percent contribution of each reaction to total initiation/termination are divided into two periods (10:00 to 14:00 and 14:00 to 18:00).