# Peer review of "OH and HO2 radical chemistry in a midlatitude forest: Measurements and"

_Atmospheric Chemistry and Physics, 2019_

## Referee Comment (RC1) · Anonymous Referee #1 · 17 Sep 2019

This study focuses on the analysis of the OH and HO2* radicals concentrations and total OH reactivity during the IRRONIC field campaign. The campaign was performed in a forested area characterized by high isoprene emissions and low NO concentrations. Measured radicals, which include a possibly interference-free OH radical measurement, are compared with two mechanisms (RACM2 and MCM) both with and without isomerization reactions for isoprene-RO2 as described within the LIM1 mechanism.

The paper is well written and the data are adequately presented. Though, the analysis of the results and the discussion of the findings is too limited and in the current status this reviewer is not sure it is enough for publication on ACP. Following are some general comments which could help improving the discussion.

One of the problems of this study is the lack of NO data for a large fraction of the

campaign. The authors overcome the issue by using the measured diurnal average when no NO data is available. I do not think this is a very good approach. Indeed, a much better solution is to constrain the model to the ozone and NO2 concentrations and jNO2 values and let the model calculate the NO. This is shown only for one model run (MCM 331) but should be done for all the models. Also, the modeled NO concentration should be compared with the measured one to see how well the model is able to reproduce it. This would allow for a better confidence in the models output also for days were no NO measurements are available.

It would be good to focus on the days when the measurements are complete and try and understand why there is still a discrepancy between modeled and measured OH radicals even after the inclusion of the LIM1 mechanisms. On those days it could be good to perform an experimental budget if possible. I can understand it could be difficult as the HO2* radical measurement is affected by an interference from RO2 radicals but a % of this interference is given based on laboratory studies so it should be possible to remove it. This would allow for an additional way to assess whether there is a discrepancy between the included sources of OH radicals and the total OH radical production.

Both RACM2 and MCM mechanisms are used in this study but there is no discussion about why both are used and, based on the results, which one is able to better reproduce the measured data and why. As both are used extensively within the community a better analysis of the differences between the two should be given. Also, both are implemented with the LIM1 mechanism. Is this done in the same way or are there differences? What is the reason behind the large differences in the modeled HO2* concentrations between the two mechanisms?

The total OH reactivity measurement shows that, overall, when the contribution from modelled OVOCs is included, the budget is closed. Though, this is not true for some days when still a certain fraction of OH reactivity is unexplained. It would be good to look if there were differences between these days and the ones were the OH reactivity

could be explained. . .different wind directions, different VOCs distribution, different meteorological conditions, etc. Does this missing reactivity correlate with the days where the measured OH concentration is larger than the modelled one?

In general, often it is written the agreement is good or there is better agreement. . .but there is no value reported of a ratio of model to measurement or correlation coefficient so it is not possible to really assess the correctness of these statements.

Specific comments:

Abstract: It would be good to have, in addition to percentages, also the mean concentrations of radicals and OH reactivity and what low NOx means.

Page 2, Line 12 to 17. No mention of the recent campaigns performed in China (Tan et al., 2017; Tan et al., 2018; Tan et al., 2019).

Page 5, Line 11. Is it pure NO injected to convert HO2 to OH radicals? Or why is the interference from RO2 radicals so large? Would not it make sense to reduce the NO further to reduce the interference?

Page 9, Line 1. Is the LIM or the LIM1 included in both RACM2 and MCM 3.3.1?

Page 10, Line 34. What does it mean that measurements on the 21-22 July focused on the HO2* thus OH measurements were not available? The instrument should measure OH and HO2 radicals in parallel or? How is stopping the OH measurement going to improve the measurement of HO2* radicals?

Page 16, Line 9. What does "similar OH" stands for?

Figure 5. Colors of the models are different between OH and HO2* panels.

Figure 12. Suggest to have consistency of the colors within the upper and lower panels.

References:

Tan, Z., Fuchs, H., Lu, K., Hofzumahaus, A., Bohn, B., Broch, S., Dong, H., Gomm, S.,

Häseler, R., He, L., Holland, F., Li, X., Liu, Y., Lu, S., Rohrer, F., Shao, M., Wang, B., Wang, M., Wu, Y., Zeng, L., Zhang, Y., Wahner, A., and Zhang, Y.: Radical chemistry at a rural site (Wangdu) in the North China Plain: observation and model calculations of OH, HO2 and RO2 radicals, Atmos. Chem. Phys., 17, 663-690, doi:10.5194/acp-17-663-2017, 2017.

Tan, Z., Rohrer, F., Lu, K., Ma, X., Bohn, B., Broch, S., Dong, H., Fuchs, H., Gkatzelis, G. I., Hofzumahaus, A., Holland, F., Li, X., Liu, Y., Liu, Y., Novelli, A., Shao, M., Wang, H., Wu, Y., Zeng, L., Hu, M., Kiendler-Scharr, A., Wahner, A., and Zhang, Y.: Wintertime photochemistry in Beijing: observations of ROx radical concentrations in the North China Plain during the BEST-ONE campaign, Atmos. Chem. Phys., 18, 12391-12411, doi:10.5194/acp-18-12391-2018, 2018.

Tan, Z., Lu, K., Hofzumahaus, A., Fuchs, H., Bohn, B., Holland, F., Liu, Y., Rohrer, F., Shao, M., Sun, K., Wu, Y., Zeng, L., Zhang, Y., Zou, Q., Kiendler-Scharr, A., Wahner, A., and Zhang, Y.: Experimental budgets of OH, HO2, and RO2 radicals and implications for ozone formation in the Pearl River Delta in China 2014, Atmos. Chem. Phys., 19, 7129-7150, doi:10.5194/acp-19-7129-2019, 2019.

---

## Referee Comment (RC2) · Anonymous Referee #2 · 5 Nov 2019

Review acp-2019-726 OH and HO2 radical chemistry in a midlatitude 1 forest: Measurements and model comparisons M. Lew et al.

The paper describes measurements of OH reactivity, OH and HO2* in a biogenic dominated regime with medium NO concentrations during daytime in the order of a few hundred pptv and compares the measurements with a box model study using four different model schemes. The main findings are that the model describes well the measured HO2* but underestimates the OH concentration. The most likely reason identified is a poor quality of the available NO measurements for the time periode shown. When the model is constrained by NO2 to calculate the NO concentration a better agreement of the model is found but local NO sources, independent of the photolysis of NO2 worsen the agreement as soon as the steady state assumption cannot be made anymore. The

paper also shows that the Indiana instrument successfully implemented a chemical scavenging modulation, improving the quality of the OH measurements. The quality of the dataset does not allow a detailed investigation of testing different model schemes, though it is obvious that the LIM1 based recycling reactions do provide a better agreement in the late afternoon when Isoprene is larger. The paper is well written, hPa seems to me preferable than Torr. The paper can be published after minor corrections.

P3 L11 : The statement "The extent of RO2 radical contributions during HO2 measurements in previous campaigns is unclear." Is not correct. For HUMPPA 2010 the contribution of a RO2 interference to the HO2* signal had been estimated based on H2O2 in Hens et al 2014, as well as calculated based on the PAA-PAN-HO2 system in Crowley et al. 2018 . Mallik et al. 2017 did model the internal production of OH from RO2 as well as compared it with a NO titration scheme done routinely in ambient air during the CYPHEX 2014 campaign.

P4 L29 : To what extend is double pulsing an issue, considering the volume flow, the expansion of the UV beam due to the white cell and the 10kHz repetition rate?

P4 L29 : please use SI units

P6 L28 : Unclear why the precision of the HO2* measurements is unrelated to the RO2 interference. Why would not the variability in the relative RO2 & HO2 composition translate into a variability in HO2* and therefore in an apparent precision ?

P8 L18: Please be more specific why a constant scaling factor can be used. Hansen et al. 2014 does not describe further the reason for the factor 1.4 beyond speculating about incomplete mixing or issues with the flow speed. Without knowing the fundamental reason for the factor, the application of such seems to be arbitrary.

P9 L27: Not conclusive is a 50% contribution of the background signal between 8:00 and 20:00. In the figure it seems rather in the range of 20%-300% even just for noontime. In any case I am not sure if the fractional description is of much use anyhow

as the relationship between ambient OH and chemical background OH is not clear at best. I would drop the discussion about the fractional contribution. You mention it above already that you are using a chemical scavenger method to remove ambient OH for quantification of the non-ambient OH.

P10 L5: Do you observe a correlation of the internal background signal with O3*BVOC as described in Novelli 2017 ?

P10 L12: Novelli 2014 proposed the presence sCI decomposition as reason for the internal OH.

P11 L4: What is the time periode used for calculating the average ? Did you model the non averaged time series and then average the model together with the measurements ?

P11 L14: "However, as seen in Fig. 6, ..." sentence seems to be reduntant to "If the measured interference was not subtracted from the total OH...."

P11 L27: Is there a NO source close by and to what extend is the assumption of steady state NO/NO2 justified? The floating NO leads to much better model estimates for OH, but seems to deviate as soon as the sun sets. From a model point of view, NO production in the model will follow JNO2, which decay quickly into the night and reduce the OH source from HO2+NO whereas the measured OH is significant different from 0, therefore the question, is there a still active NO source close by ?

P11 & P16 Check spelling of the name, Rohrer

P 16 L25: Mallik et al, 2017 like Mao found a decent agreement of modeled OH and measured OH only when the interference, determined by a chemical modulation technique had been taken into account. I would be careful with a generalization, the instrument by the Leeds and Juelich group seem to be not as much as sensitive to the interference as the PennState/Indiana/Mainz & Lile group. The most striking difference is the use of a multipass cell vs. single beam cell.

---

## Author Comment (AC1) · 13 Jan 2020

We would like to thank the reviewers for their efforts in reviewing this manuscript, and we feel that the manuscript is much stronger with the suggested changes. Below are detailed responses to their comments, which are highlighted in italics.

Reviewer #1

This study focuses on the analysis of the OH and HO2* radicals concentrations and total OH reactivity during the IRRONIC field campaign. The campaign was performed in a forested area characterized by high isoprene emissions and low NO concentrations. Measured radicals, which include a possibly interference-free OH radical measurement, are compared with two mechanisms (RACM2 and MCM) both with and without isomerization reactions for isoprene-RO2 as described within the LIM1 mechanism.

The paper is well written and the data are adequately presented. Though, the analysis of the results and the discussion of the findings is too limited and in the current status this reviewer is not sure it is enough for publication on ACP. Following are some general comments which could help improving the discussion.

One of the problems of this study is the lack of NO data for a large fraction of the campaign. The authors overcome the issue by using the measured diurnal average when no NO data is available. I do not think this is a very good approach. Indeed, a much better solution is to constrain the model to the ozone and NO2 concentrations and jNO2 values and let the model calculate the NO. This is shown only for one model run (MCM 331) but should be done for all the models. Also, the modeled NO concentration should be compared with the measured one to see how well the model is able to reproduce it. This would allow for a better confidence in the models output also for days were no NO measurements are available.

*We agree that constraining the model to the measured ozone, NO2, and jNO2 and allowing the model to calculate NO is an alternative approach to addressing the lack of NO measurements in this study. However, the measurements suggest that deviations from the ozone photostationary state were significant at this site, implying that the concentrations of peroxy radicals were high enough to significantly impact the concentration of NO. In addition, the location of the site relatively close to NO sources from transportation as well as soil emissions may also impact the NO/NO2 ratio, as pointed out by Reviewer #2. For these reasons, we chose to constrain the model to the measured diurnal averaged NO to predict the radical concentrations rather than allowing the model to calculate the NO. We have clarified this in the revised manuscript on page 9:*

*"Zero-dimensional models cannot explicitly account for emissions, and NO is emitted both by vehicles on the nearby highway 1 km to the Southwest and by soil. Such local perturbations to the NOx-O3-radical chemistry necessitate using constrained measurements of NO, NO2, and O3."*

*However, we have included additional model runs with NO calculated by the model as suggested by the reviewer. Figure 6 in the revised manuscript now includes results from the MCM 3.2*

*model in addition to the MCM3.3.1 results. For clarity, the NO unconstrained RACM2 and RACM2-LIM1 results are included in an additional plot in the Supporting Information.*

*On days when NO was measured, the modeled calculated NO overpredicted the measured NO during the day, and underpredicted it in the morning and at night, suggesting that the assumption of an NO/NO2 steady-state may not be justified. An additional plot illustrating the inability of the model to reproduce the measured NO is also included in the Supporting Information.*

*We have added the following to the revised manuscript addressing these points on page 12:*

*"However, the assumption that NO is in steady-state may not be justified given the location of the site near NO sources from transportation as well as the potential influence of soil emissions (Molina-Herrera et al., 2017). In addition, the measurements suggest that deviations from the ozone photostationary state were significant at this site, implying that the concentrations of peroxy radicals were high enough to significantly impact the concentration of NO. On the days when NO was measured, the models overpredicted the NO measurements by a factor of approximately 2-4 during the day, and underpredicted the measurements in the morning and evening (Fig. S3). This underprediction of the measured NO in the morning and evening may reflect active NO sources from soil and transportation emissions, and could explain why the NO unconstrained model underpredicts the concentration of OH in the afternoon."*

It would be good to focus on the days when the measurements are complete and try and understand why there is still a discrepancy between modeled and measured OH radicals even after the inclusion of the LIM1 mechanisms. On those days it could be good to perform an experimental budget if possible. I can understand it could be difficult as the HO2* radical measurement is affected by an interference from RO2 radicals but a % of this interference is given based on laboratory studies so it should be possible to remove it. This would allow for an additional way to assess whether there is a discrepancy between the included sources of OH radicals and the total OH radical production.

*Unfortunately there were only a few days during the campaign when the measurements were complete. As discussed in the manuscript, the models including the LIM1 are in better agreement with the measured OH and HO2* concentrations on these days, illustrating the ability of the model to reproduce the measurements when NO was measured simultaneously. For 20 July, the RACM2-LIM1 and MCM 3.3.1 models predict a maximum concentration of OH that are within 30% of the measured concentration, in better agreement than the models without the LIM1 mechanism. We have clarified this in the revised manuscript on page 12 and included an additional figure in the Supporting Information highlighting the agreement of the LIM1 models with the measurements on this day:*

*"For the days at the end of the campaign where there was significant overlap between the measurements of OH and NO, the model results are in better agreement during these days (20*

*and 24 July) (Fig. 5). On 20 July, the RACM2-LIM1 and MCM 3.3.1 models predict maximum OH concentrations that are within 30% of the measured OH on 20 July (Fig. S1).*

*As pointed out by the reviewer, an experimental budget for the days where the measurements are complete can provide some information regarding the source of discrepancies with the model. An experimental OH radical budget analysis for 20 July suggests that the measured OH production is less than total OH loss. However, including the modeled OH production from the LIM1 mechanism brings the total OH production into reasonable agreement with total OH loss on this day, with the measured OH production rates (including the modeled LIM1 contribution) within 30% of the measured OH loss rate, consistent with the ability of the LIM1 mechanisms to reproduce the maximum measured OH concentration on this day. This is discussed in the revised manuscript on page 15 and the experimental budget is included in the Supporting Material:*

*"An experimental radical budget for 20 July when the measurements were complete suggests that the total measured OH production rate is nearly balanced by the total OH loss rate calculated by the concentration of individual sinks and the loss rate based on the measured total OH reactivity to within approximately 30% (Fig. S8), consistent with the agreement between the measured and modeled OH on this day as discussed above. For simplicity, the measured HO2\* was used to calculate the rate of OH production from the HO2 + NO reaction and as a result the measured production rate represents an upper limit to the overall OH production rate. Thus, the difference between production and loss may be greater than illustrated in this figure, but is still likely to be within the combined uncertainties of all the measurements (for example 38% (2σ) for OH and for HO2), similar to that observed previously (Tan et al., 2019)."*

Both RACM2 and MCM mechanisms are used in this study but there is no discussion about why both are used and, based on the results, which one is able to better reproduce the measured data and why. As both are used extensively within the community a better analysis of the differences between the two should be given. Also, both are implemented with the LIM1 mechanism. Is this done in the same way or are there differences? What is the reason behind the large differences in the modeled HO2\* concentrations between the two mechanisms?

*As indicated by the reviewer, we chose to use both RACM and MCM mechanisms to model these results given that both are used extensively within the community. While the MCM mechanism is a more explicit mechanism expected to better reproduce complex systems, the lumped RACM mechanism provides a simpler radical budget analysis. We have clarified the reason for using these mechanisms on page 9 of the revised manuscript, in addition to clarifying how the LIM1 mechanism is incorporated in each:*

*"While the MCM model provides a near-explicit chemical mechanism and is expected to better represent complex chemical atmospheres, the lumped RACM mechanism is easier to use in radical budget calculations. The isoprene oxidation mechanism in RACM2 was updated as described in Tan et al. (2017) to include the Leuven Isoprene Mechanism (LIM1) originally proposed by Peeters, et al. (2009) involving peroxy radical isomerization reactions leading to additional HOx radical production, and includes the LIM1 updated bulk RO2 reactions*

*described in Peeters et al. (2014). The addition also includes a revision of the chemistry of first-generation isoprene oxidation products, including methyl vinyl ketone (MVK), methacrolein (MACR), and isoprene hydroperoxides (ISHP) (Tan et al., 2017). In addition, the ambient measurements were also modeled with version 3.3.1 of the Master Chemical Mechanism (MCM). In comparison to MCM 3.2, MCM 3.3.1 includes an updated isoprene oxidation mechanism based on the LIM1 mechanism resulting in HOx recycling from peroxy radical H-shift isomerization reactions (Jenkin et al., 2015)."*

*During an analysis in response to the reviewer noting the difference between the modeled HO2\* by the two mechanisms, we found a discrepancy in the model results illustrated in Figure 6. Correcting this discrepancy resulted in better agreement in the modeled HO2\*concentrations by the two mechanisms, with predicted maximum concentrations agreeing to within 10%. This has been clarified in the revised manuscript and the corresponding figures and discussion has been updated.*

The total OH reactivity measurement shows that, overall, when the contribution from modelled OVOCs is included, the budget is closed. Though, this is not true for some days when still a certain fraction of OH reactivity is unexplained. It would be good to look if there were differences between these days and the ones were the OH reactivity could be explained…different wind directions, different VOCs distribution, different meteorological conditions, etc. Does this missing reactivity correlate with the days where the measured OH concentration is larger than the modelled one?

*We have done an extensive analysis of the missing reactivity measured during this campaign, including analyzing different wind directions, velocity, trajectories, meteorological conditions, etc. and have yet to find an explanation for the missing reactivity. We are continuing this analysis and plan to do additional measurements in the future. This has been clarified on page 15 of the revised manuscript:*

*"The reason for this discrepancy is unclear, as the missing reactivity on this day did not appear to correlate with changes in wind speed, direction, trajectory, or meteorological conditions, but may indicate the presence of additional unmeasured emissions or oxidation products not accounted for by the model. Additional measurements and analyses will be necessary to determine the source of the missing reactivity."*

In general, often it is written the agreement is good or there is better agreement… but there is no value reported of a ratio of model to measurement or correlation coefficient so it is not possible to really assess the correctness of these statements.

*We have attempted to include more quantifiable comparisons in the revised manuscript as suggested.*

Specific comments:

Abstract: It would be good to have, in addition to percentages, also the mean concentrations of radicals and OH reactivity and what low NOx means.

*These suggestions have been added to the abstract.*

Page 2, Line 12 to 17. No mention of the recent campaigns performed in China (Tan et al., 2017; Tan et al., 2018; Tan et al., 2019).

*We have added these references as suggested.*

Page 5, Line 11. Is it pure NO injected to convert HO2 to OH radicals? Or why is the interference from RO2 radicals so large? Would not it make sense to reduce the NO further to reduce the interference?

*One of the goals of this study was an instrumental intercomparison of peroxy radical measurements by the IU-FAGE instrument with the HO2+RO2 measurements by the Drexel University Ethane – Nitric Oxide Chemical Amplifier (ECHAMP) instrument. In order to provide a useful intercomparison, high concentrations of NO (10% in N2) were deliberately added to the IU-FAGE instrument to allow for efficient conversion of isoprene peroxy radicals in addition to HO2. Given that the total peroxy radical concentrations at this site were primarily HO2 and isoprene peroxy radicals, the resulting HO2\* measurements were found to be similar to the total HO2+RO2 measurements by the ECHAMP instrument. The results of this intercomparison are summarized in a separate publication (Kundu et al., 2019). This has been clarified on page 7 of the revised manuscript:*

*"A high concentration of NO leading to a high conversion efficiency of isoprene-based peroxy radicals to HO2 was used throughout the study to provide a useful intercomparison of the IU-FAGE HO2\* measurements with the RO2+HO2 measurements by the Drexel University Ethane – Nitric Oxide Chemical Amplifier (ECHAMP) instrument (Kundu et al., 2019), as HO2 and isoprene-based peroxy radicals accounted for approximately 70% of the total peroxy radicals at this site (see below)."*

Page 9, Line 1. Is the LIM or the LIM1 included in both RACM2 and MCM 3.3.1?

*The updated RACM2 mechanism (RACM2-LIM1) described in Tan et al. (2017) includes updates to the LIM rates as described in Peeters et al., 2014. This has been clarified in the revised manuscript. The MCM 3.3.1 mechanism also includes the updated LIM1 mechanism as described in Jenkin et al., (2015) and this has been clarified in the revised manuscript, as noted above.*

Page 10, Line 34. What does it mean that measurements on the 21-22 July focused on the HO2* thus OH measurements were not available? The instrument should measure OH and HO2 radicals in parallel or? How is stopping the OH measurement going to improve the measurement of HO2* radicals?

*The IU-FAGE instrument is composed of a single detection axis for measuring both OH and HO2, and as a result cannot measure OH and HO2 simultaneously. This has been clarified on page 4 of the revised manuscript:*

*"The Indiana University LIF-FAGE instrument (IU-FAGE) has been described in detail previously and consists of a single axis for alternating measurements of OH and HO2 or HO2* (Dusanter et al., 2009a Griffith et al., 2013; 2016)."*

*For most of the campaign, NO was added for 30 seconds every 30 minutes to measure HO2*, while OH was measured during the remaining time. As part of the peroxy radical intercomparison with the Drexel University ECHAMP instrument, NO was added continuously on 21-22 July to allow for higher time resolved measurements of HO2*. Because NO was added continuously during this time, no OH measurements were made on these days. This has been clarified on page 11 of the revised manuscript:*

*"Measurements on 21-22 July focused on measurements of HO2* as part of the peroxy radical informal instrumental intercomparison (Kundu et al., 2019), with NO added continuously to the detection cell to provide measurements with a higher time resolution. Thus OH measurements were not conducted on these days."*

Page 16, Line 9. What does "similar OH" stands for?

*This has been revised to* *"similar concentrations of OH have been observed at this site…"*

Figure 5. Colors of the models are different between OH and HO2* panels.

*We have corrected the panels so that the color of the model results are consistent.*

Figure 12. Suggest to have consistency of the colors within the upper and lower panels.

*We have revised the colors as suggested.*

---

## Author Comment (AC2) · 13 Jan 2020

We would like to thank the reviewers for their efforts in reviewing this manuscript, and we feel that the manuscript is much stronger with the suggested changes. Below are detailed responses to their comments, which are highlighted in italics.

Reviewer #2

The paper describes measurements of OH reactivity, OH and HO2* in a biogenic dominated regime with medium NO concentrations during daytime in the order of a few hundred pptv and compares the measurements with a box model study using four different model schemes. The main findings are that the model describes well the measured HO2* but underestimates the OH concentration. The most likely reason identified is a poor quality of the available NO measurements for the time periode shown. When the model is constrained by NO2 to calculate the NO concentration a better agreement of the model is found but local NO sources, independent of the photolysis of NO2 worsen the agreement as soon as the steady state assumption cannot be made anymore. The paper also shows that the Indiana instrument successfully implemented a chemical scavenging modulation, improving the quality of the OH measurements. The quality of the dataset does not allow a detailed investigation of testing different model schemes, though it is obvious that the LIM1 based recycling reactions do provide a better agreement in the late afternoon when Isoprene is larger. The paper is well written, hPa seems to me preferable than Torr. The paper can be published after minor corrections.

P3 L11 : The statement "The extent of RO2 radical contributions during HO2 measurements in previous campaigns is unclear." Is not correct. For HUMPPA 2010 the contribution of a RO2 interference to the HO2* signal had been estimated based on H2O2 in Hens et al 2014, as well as calculated based on the PAA-PAN-HO2 system in Crowley et al. 2018 . Mallik et al. 2017 did model the internal production of OH from RO2 as well as compared it with a NO titration scheme done routinely in ambient air during the CYPHEX 2014 campaign.

*We have clarified on page 3 of the revised manuscript that while there are several campaigns where the extent of any RO2 interference on the HO2 measurements has been estimated and accounted for (and have included the noted references in the revised manuscript), there are many previous studies which the contribution of a RO2 interference in the HO2 measurements is not known:*

*"The degree to which the RO2 species can interfere with HO2 measurements has been quantified through several laboratory experiments (Fuchs et al., 2011; Whalley et al., 2013; Lew et al., 2018) and estimated in some field studies (Hens et al., 2014; Crowley et al., 2018; Mallik et al., 2018). However, the extent of RO2 radical contributions during HO2 measurements in many of the campaigns mentioned above is unclear."*

P4 L29 : To what extend is double pulsing an issue, considering the volume flow, the expansion of the UV beam due to the white cell and the 10kHz repetition rate?

*The instrument design and configuration is similar to that described in Stevens et al. (1994). The flow velocity in the region of the White cell is likely greater than the 50 m s-1 required to prevent double-pulsing of the air stream. However, any laser-generated OH is measured and accounted for through the interference measurements.*

P4 L29 : please use SI units

*We have converted to SI units as requested.*

P6 L28 : Unclear why the precision of the HO2* measurements is unrelated to the RO2 interference. Why would not the variability in the relative RO2 & HO2 composition translate into a variability in HO2* and therefore in an apparent precision?

*We have clarified on page 7 of the revised manuscript that the instrumental precision of the measurements is primarily related to the variation in the background signal due to laser scatter and detector noise in the detection cell:*

*"The instrumental precision for the HO2* measurement based on the variability of the background signal due to laser scatter and detector noise results in a limit of detection for HO2* during this campaign of 7 × 107 cm-3 for a 30 second average (S/N =1)."*

P8 L18: Please be more specific why a constant scaling factor can be used. Hansen et al. 2014 does not describe further the reason for the factor 1.4 beyond speculating about incomplete mixing or issues with the flow speed. Without knowing the fundamental reason for the factor, the application of such seems to be arbitrary.

*The constant scaling factor is derived from reproducible OH reactivity measurements in the laboratory of a variety of compounds, including butane, isoprene, and propane, which were all found to be a factor of 1.4 lower than the calculated reactivity. This has been clarified on page 8 of the revised manuscript:*

*"Laboratory measurements of the reactivity of several VOCs with well-known rate constants, including butane, isoprene, and propane showed that the OH reactivity measurements for these compounds were on average 30% lower than calculated when the measured velocity of the turbulent core is used to determine the reaction time. This consistent underestimation of the OH reactivity is likely due to either incomplete mixing of the reactants or a systematic underestimation of the reaction time (Hansen et al, 2014)."*

P9 L27: Not conclusive is a 50% contribution of the background signal between 8:00 and 20:00. In the figure it seems rather in the range of 20%-300% even just for noontime. In any case I am not sure if the fractional description is of much use anyhow as the relationship between ambient OH and chemical background OH is not clear at best. I would drop the discussion about the

fractional contribution. You mention it above already that you are using a chemical scavenger method to remove ambient OH for quantification of the non-ambient OH.

*We have removed this discussion as suggested.*

P10 L5: Do you observe a correlation of the internal background signal with O3*BVOC as described in Novelli 2017?

*The unknown interference was found to increase with the product of ozone and BVOC concentrations, similar to that found in Novelli et al., 2017, although the correlation was not statistically significant. This has been clarified on page 10 of the revised manuscript:*

*"This result is also consistent with the measurements of Novelli et al. (2017), who found that their observed interference correlated with the product of ozone and biogenic VOC concentrations, although the correlation in the present study was not statistically significant."*

P10 L12: Novelli 2014 proposed the presence sCI decomposition as reason for the internal OH.

*We have added this reference as suggested.*

P11 L4: What is the time periode used for calculating the average? Did you model the non averaged time series and then average the model together with the measurements?

*For the diurnal average shown in Fig. 6, the 15-min OH measurements with and without the interference were averaged into 1 hour bins. This has been clarified in the revised manuscript. The model results in this figure represent a model constrained by the diurnal average of the daily measurement shown in in Fig. 2, and is similar to the diurnal average of the model results. However, we did find a discrepancy in the model results previously illustrated in Figure 6. This has been corrected in the revised manuscript, and all the relevant figures have been updated.*

P11 L14: "However, as seen in Fig. 6, . . ." sentence seems to be reduntant to "If the measured interference was not subtracted from the total OH. . .."

*We have removed this sentence as suggested.*

P11 L27: Is there a NO source close by and to what extend is the assumption of steady state NO/NO2 justified? The floating NO leads to much better model estimates for OH, but seems to deviate as soon as the sun sets. From a model point of view, NO production in the model will follow JNO2, which decay quickly into the night and reduce the OH source from HO2+NO

whereas the measured OH is significant different from 0, therefore the question, is there a still active NO source close by?

*As pointed out by the reviewer, the models tend to underpredict the measured nighttime NO, suggesting that the site may be influenced by transportation by the nearby highway or perhaps more importantly local emissions of NO from soil. These active NO source likely impact the measured OH concentration and could explain the discrepancy between the measured and modeled OH concentrations in the late afternoon and early evening. This has been clarified on page 12 of the revised manuscript:*

*"However, the assumption that NO is in steady-state may not be justified given the location of the site near NO sources from transportation as well as the potential influence of soil emissions (Molina-Herrera et al., 2017). In addition, the measurements suggest that deviations from the ozone photostationary state were significant at this site, implying that the concentrations of peroxy radicals were high enough to significantly impact the concentration of NO. On the days when NO was measured, the models overpredicted the NO measurements by a factor of approximately 2-4 during the day, and underpredicted the measurements in the morning and evening (Fig. S3). This underprediction of the measured NO in the morning and evening may reflect active NO sources from soil and transportation emissions, and could explain why the NO unconstrained model underpredicts the concentration of OH in the afternoon."*

P11 & P16 Check spelling of the name, Rohrer

*This typo has been corrected.*

P 16 L25: Mallik et al, 2017 like Mao found a decent agreement of modeled OH and measured OH only when the interference, determined by a chemical modulation technique had been taken into account. I would be careful with a generalization, the instrument by the Leeds and Juelich group seem to be not as much as sensitive to the interference as the PennState/Indiana/Mainz & Lile group. The most striking difference is the use of a multipass cell vs. single beam cell.

*We have included the Mallik et al., 2017 reference as suggested. We agree that the observed interference may not be similar for all instruments, and have revised the statement accordingly on page 17 of the revised manuscript:*

*"However, it is clear that if the measured interference was not taken into account, the apparent OH concentrations would have been a factor of 5 greater than predicted by the model mechanisms, comparable to previous measurements under low NOx and high isoprene conditions (Rhorer et al., 2014). These results are similar to that reported by Mao et al. (2012) and Mallik et al. (2017) who found good agreement between their OH measurements and model predictions when measured interferences were taken into account. However, because of differences in instrument design (geometry, cell pressure, flow, etc.) these interferences may not significantly impact other LIF-FAGE instruments. However, future OH measurements using the*

*LIF-FAGE technique should include methods to quantify potential instrumental artifacts even if they are insignificant, to demonstrate that the measurements are free from interferences."*

---

## Editor Decision (ED1)

The revised paper has been greatly improved and the comments of the reviewers and the editor have been adequately answered. Before the paper can be published, some minor changes are needed for clarification.

Minor changes

- The abstract should be more specific and mention which chemical mechanisms have been tested and which version has given the best agreement with the interference corrected OH data. A quantitative statement should be made how much the modeled OH is increased by the LIM1 chemistry. Important boundary conditions (temperature, isoprene and NO concentrations) should be stated.
- Section 2.4: the difference between the two used isoprene mechanisms should be more precisely explained. The implementation in RACM2 as described in Tan et al. (2017) does not include the explicit LIM1 mechanism, but uses bulk reaction rates for the isoprene RO2 isomerization via 1,6-H shift derived as parametrizations from LIM1 by Peters et al. (2014). Contrary, MCM v3.3.1 contains the full LIM1 mechanism with a description of the equilibrium between different isoprene RO2 isomers and isomerization by H shift reactions of specific isomers (Jenkin et al., 2015). Following a reviewer recommendation by Peeters et al. (2015), Jenkin et al. (2015) adjusted the rate coefficients for equilibration and H shift reactions to match preliminary experimental results by Crounse et al. (2014). Compared to the original LIM1, these changes mean a reduction of the effective bulk rate coefficient for the 1,6-H shift by about a factor of 5.
- Page 12, line 22-23. Please quantify the difference between the model runs with and without LIM1 chemistry. The corresponding levels of NO from 9 to 17 EDT should be mentioned in this context.
- Page 12, line 22-26. The comparison between the measured and modeled OH concentrations (Figure 5, upper right panel) needs to be done more careful. The statement "... with the RACM2-LIM1 results within 30% of the measured concentrations during the day (9:00-17:00 EDT)" is not true at 14h, where the RACM-LIM1 model and the measurements are numerically different by a factor of 1.8. Probably, this discrepancy is explainable by the total uncertainty of the measured data point (OH calibration error of 18%, plus statistical error bar, plus additional uncertainty from the interference correction mentioned on the same page) and the model uncertainty of 30%.
- Page 12, line 27-29. The comparison with results by Novelli et al. is only meaningful, if the NO concentrations are comparable. To which NO concentration does the factor of 1.4 refer? The description of the improved mechanism in Novelli et al. is not entirely correct. The improvement is obtained by using the fast rate coefficients for the RO2 equilibrium from MCM v3.3.1 and the fast 1,6-H-shift rate coefficients from the Caltech mechanism (Wennberg et al., 2018). This combination causes a faster production of HPALD and di-HPCARP-RO2, both of which are likely important for the OH regeneration. The overall mechanism has a bulk rate coefficient for the 1,6-H shift which is close to the value in the original LIM1.
- Figure 6 and 8: which measurement days are averaged in the diurnal profiles?
- Figure 9: what is the meaning of a median diurnal average for a single day?
- Figure 10 and 11: which components are meant by the label 'others' in the legends. Please give a short explanation in the figure captions.

---

## Author Response (AR2)

Dear Dr. Hofzumahaus,

Thank you for your review of our revised manuscript and the return of the reviewer's comments. We have revised the manuscript as you and the reviewers have suggested, and feel that the manuscript is much improved. Below are our responses to yours and the reviewer's comments, highlighted in italics, with changes to the manuscript highlighted in red.

Editor's Comments

The revised paper has been reviewed again by two referees. Referee #1 proposes to reject the paper mainly because of insufficient availability and quality of data, while Referee #3 recommends publication with minor revisions. The requested revisions, however, are substantial and concern the lack of NO data needed for the interpretation of the radical chemistry.

I think that both referees make some good points. I share the view of Referee #3 that the paper contains valuable information that is worth to be published. It is of general interest that a significant OH interference of roughly a factor of 2 has been observed in the Indiana University LIF-FAGE instrument in a forest environment by using a chemical modulation technique. This information is relevant because similar types of instruments have reported in the past unexplained high OH concentrations in environments with high isoprene and low NO concentrations. The present work also demonstrates the usefulness of the chemical modulation method that was introduced by Mao et al. (2012). Furthermore, the well characterized data set for HO2*, which is essentially the sum of HO2 and isoprene peroxy radicals, and the measured OH reactivities are also of interest in combination with interference-corrected OH data.

I share the view of Referee #1 and #3 that the use of a mean diurnal NO profile for the intepretation of the complete radical data set is problematic. There are large gaps in the measured time series of OH and NO, which show little temporal overlap between the records. The paper presents all arguments why it is not reasonable to use the averaged diurnal NO profile for OH model calculations which are then compared to the measured OH. First, NOx levels were much higher in the first period of the campaign, when most of the OH data were measured, compared to the second period, when most NO data were recorded. Second, missing NO could not be calculated from the available NO2 measurements, because the photostationary state was strongly perturbed by local NO sources. In such an environment, the lack of measured NO data prevents firm conclusions from comparisons between modelled and measured OH, as the production of OH is dominated by the reaction of HO2+NO. Following the recommendation of Referee #3 and in agreement with comments by Referee #1, I suggest that the authors revise their paper and restrict the model interpretation of OH and HO2* only to those periods when NO and NO2 are simultaneously available. Though this means that only a few days can be used for comparison between model and measurements, the conclusions are expected to be significantly more robust than in the current paper version. This may provide better constraints to answer the important question, whether we are still lacking unknown OH (or HO2) radical sources in current atmospheric chemical mechanisms. When your revise the manuscript, the suggestions by the referees should be taken into account. In addition, I also have a few specific suggestions that should be considered.

*As recommended, we have revised the paper and focused the discussion of the model interpretation to the few days where both OH and NO were available. The diurnal average of this data with the model results has been included as a new figure in the revised manuscript. We have removed the model results for days when NO measurements were not available, and removed the discussion of the model results with NO unconstrained. The discussion of the model results is now focused on these days, and the relevant figures*

*have been updated. While the main conclusions of the paper have not changed, we have added the a discussion of the results of this new analysis to the paper.*

Editor's suggestions

• The magnitude of the measured OH interference should be mentioned in the abstract.

*We have included the magnitude of the average interference in the abstract as suggested.*

"Using an OH chemical scavenger technique, the study revealed the presence of an interference with the LIF-FAGE measurements of OH that increased with both ambient concentrations of ozone and temperature, with an average daytime maximum equivalent OH concentration of approximately $5\times10^6$ $cm^{-3}$."

• The quoted literature on the chemistry of isoprene is not up-to-date. Since the LIM1 mechanism was published by Peeters et al. (2014), there have been new studies presented by Peeters (2015), Teng et al. (2017), Wennberg (2018), Berndt et al. (2019), Müller et al. (2019) and Novelli et al. (2020). For references and review of the differences in the mechanisms, see the recent paper by Novelli et al. (2020). It should be noted that the LIM1 mechanism and the isoprene mechanism in MCM v3.3.1, which are both used in the present study, are not the same, as they use different rate coefficients for the LIM1 chemistry. This should be clarified on page 9, line 12-13.

*We have expanded the discussion of isoprene chemistry, updating the references as suggested. We have also clarified the differences in the rate coefficients for the bulk isomerization rate constants between the LIM1 mechanism and MCM 3.3.1 on page 9 of the revised manuscript as suggested.*

"The bulk peroxy radical isomerization rate constants in MCM 3.3.1 are based on the recommendations of Peeters (2015), which are approximately a factor of 5 lower than the original LIM1 recommended rates (Peeters, 2015; Novelli et al., 2020) in order to bring the model predictions into better agreement with experimental measurements of the production of HPALDs and other products (Crounse et al., 2011; Teng et al., 2017; Wennberg et al., 2018; Berndt et al., 2019)."

• Page 5, line 7. Does the laser power (0.5 - 4.4 mW) apply to a single laser beam, or to overlapping laser beams?

*We have clarified that this laser power reflects the power entering the sampling cell and does not represent the power density due to overlapping beams inside the detection cell on page 5 of the revised manuscript.*

"For this campaign, the laser power entering the sampling cell ranged from 0.5 to 4.4 mW and was monitored using a photodiode at the exit of the White cell. This does not reflect the laser power density inside the detection cell due to overlap of the beams in the multipass configuration."

• In Figure 5, a considerable number of interference-corrected OH data points have negative values that are more than 3sigma less than zero. Why? Is it possible that the chemical modulation is overcorrecting the interference or is the precision of the OH data worse than indicated by the error bars? This point is also addressed by Referee #1 who is wondering about the negative OH concentration in the afternoon of July 20 (Figure S1). In Figure S8, this data has apparently been discarded and replaced by estimated values. The figure caption only mentions "Concentrations of OH in the afternoon were estimated due to the poor precision of the data." Please explain the reason and how the estimate was done.

*The large negative values (which occurred primarily at night) reflect the fact that the interference was not measured simultaneously as the OH measurements. Because the interference during these times was much larger than the ambient OH signal, large variations in the interference between measurement cycles often resulted in additional scattering in the net ambient OH signals derived from subtraction of the interference, which sometimes exceeded the precision of individual measurements (OH plus interference and the interference). This has been clarified on page 11 of the revised manuscript.*

"Because the interference was not measured simultaneously as the ambient OH measurements, subtraction of the measured interference often resulted in both apparent negative concentrations as well as large positive concentrations at night. These large positive and negative values reflect the fact that the nighttime measurements of the interference was much greater than the ambient OH signals and was highly variable between measurement cycles (Fig. 3), resulting in an ambient OH measurement uncertainty that was sometimes larger than the precision calculated from a quadratic propagation of the errors associated with the individual measurements of the ambient OH plus the interference and the interference alone."

*The experimental radical budget in the supplement (formally Figure S8) has been revised to include an average of all days that include NO measurements, and with the additional data, estimation of missing OH concentrations for this particular day is no longer necessary.*

• When the measured OH concentrations from the IRRONIC campaign are compared with different models, I recommend to compare the outcome with the results derived by Novelli et al. (2020) from chamber experiments. She investigated the photooxidation of isoprene by OH in synthetic air at ambient conditions and comparable low NO concentrations. The measured OH is compared with predictions by different chemical mechanisms including MCM v3.3.1 (Figure 5, 6) and LIM1 (Figure S5), which are also used in the present work.

*As suggested, we have included a comparison of the results presented here with the results of Novelli et al. (2020), who also compared their measurements in the SAPHIR chamber and found that the MCM 3.3.1 mechanism underpredicted the OH measurements. They found that a model incorporating bulk isoprene peroxy radical isomerization rates similar to that used in the LIM1 mechanism were in better agreement with the measurements, similar to that reported here. This discussion has been added to page 12 of the revised manuscript as well as in the Summary.*

"Including versions of the LIM1 mechanism for HOx regeneration in both the MCM (3.3.1) and RACM2 (RACM2-LIM1) models result in higher modeled daytime concentrations of OH compared to the base MCM 3.2 and RACM2 mechanisms, with the RACM2-LIM1 results in better agreement and within 30% of the measured concentrations during the day (9:00-17:00 EDT) (Fig. 5), while the MCM 3.3.1 model underpredicted the measurements during this period by approximately a factor of 2. These results are similar to that found by Novelli et al. (2020), who found that the MCM 3.3.1 underpredicted measurements of OH by a factor of approximately 1.4 during isoprene oxidation experiments in the SAPHIR chamber. The measured OH concentrations could be reproduced using a model that increased the yield of HPALD in the oxidation mechanism, resulting in an effective bulk isoprene peroxy radical isomerization rate similar to that in the original LIM1 mechanism (Novelli et al., 2020). These larger bulk peroxy radical isomerization rates are incorporated into the RACM2-LIM1 mechanism used in the present study (Tan et al., 2017), leading to the higher modeled radical concentrations compared to the RACM2 and MCM model results shown in Fig. 5."

• Figure S8. I agree with Referee #1. Since HO2* contains a well characterized fraction of isoprene peroxy radicals, I suggest to use interference-corrected HO2 concentrations for the OH budget analysis.

*While the contribution of isoprene peroxy radicals to HO2\* has been quantified in the lab, correcting the ambient measurements for the interference requires a knowledge of the ambient concentration of isoprene peroxy radicals, which for this study can only come from model predictions. For the experimental radical budget shown in the supplement, we chose to use the experimentally measured HO2\* concentrations rather than corrected HO2 concentrations based on model predictions. However, for this revision we have included the estimated contribution of the HO2+NO reaction to the experimental OH radical budget based on estimates of the ambient HO2 concentration from the measured HO2\* concentrations and the modeled HO2/HO2\* ratio.*

• Is there any explanation for the relative high midnight isoprene values (1ppbv)? For comparison, Tan et al. (JGR, 2001) report 10 times lower nocturnal isoprene concentrations in an isoprene emitting forest in summer 1998.

*The high nighttime mixing ratios of isoprene observed at this site are likely due to the fact that the measurements were conducted near the surface and below the forest canopy, in contrast to the measurements by Tan et al. (2001) which were sampled approximately 10-m above the forest canopy on top of the 31-m PROPHET tower. Because the measurements in this study were conducted below the forest canopy, vertical stratification likely resulted in higher concentrations of isoprene trapped near the surface during several nights. This has been clarified on page 10 of the revised manuscript.*

"The relatively high nighttime mixing ratios often observed at this site are likely due to the fact that the measurements were made below the forest canopy and relatively close to the surface. As a result, vertical stratification likely resulted in higher concentrations of isoprene near the surface during several nights, similar to other measurements of biogenic VOCs below the forest canopy (Bsaibes, et al., 2020)."

• The paper uses EDT time. When was local noon?

*Local solar noon at this site occurred at approximately 13:52 EDT. This has been clarified on page 4 of the revised manuscript.*

• Page 17, line 25: Typo. It must read "Rohrer".

*This typo has been corrected.*

• Table 1. Add footnote explaining HO2\*.

*We have added a footnote defining HO2\* to the table, as suggested.*

Reviewer 2

I have reviewed the manuscript in view of the interactive public discussion, looking at the reports of the reviewers, the responses and the revised MS and supplement. I am happy that the authors have in general responded satisfactorily to the reviews, and made appropriate changes to the MS. There are a few areas though where some further clarifications are needed.

The lack of NO measurements during the campaign, and the probable local sources of NOx (soil, traffic) causing the stated deviation from steady-state (also from large concentrations of peroxy radicals), means that alternate methods of estimating NO during these periods, e.g. from NO2 measurements, seemingly are not easy to implement. This is clearly stated in the paper. Therefore focussing on the periods were both NO and NO2 measurements are available is sensible and being careful that average diel profiles of model calculations and therefore conclusions regarding the level of agreement are only generated from periods where NO data are available to constrain the model. There are NO data though for significant periods, which makes the model comparison a useful and valid thing to do. Figure 1 and Figure S3 certainly supports the idea that NO is not in photostationary state with NO2, and there are local sources (soil, traffic) – which is now clearly stated in the revised MS.

*As mentioned above, we have revised the discussion of the model comparison to the days when NO was measured, as suggested.*

Page 7 (revised MS) lines 1-6, it is not clear why just HO2* was measured? I realise this means that a comparison with the ethane PERCA instrument (some of HO2+RO2) could be performed (the latter measuring HO2+mainly isoprene RO2 which forms an interference in HO2), and this was the subject of a previous publication, but was the NO ever switched to a low value some of the time to enable HO2 measurements periodically to compare with modelled HO2?

*Unfortunately, because a major goal of the campaign was the intercomparison with the ECHAMP instrument, the NO concentration added to the instrument was not reduced during the campaign to measure HO2 for comparison with the model.*

The OH interference, although sometimes significant, has been characterised for this instrument, and so there should be confidence that the OH measurements from OHchem are accurate. Also, it is fairly likely that HO2* is the sum of HO2 and some fraction of isoprene peroxy radicals (which is characterised in the lab and with this RO2 being the main RO2 species). From this point of view the measurement time-series of radicals of OH and HO2* is valuable, and as the data have not been over interpreted and sensible comparisons with the model have been made (i.e. where NO measurements are available to constrain the model), then this is a worthwhile contribution.

Some subjective phrases have been made more quantitative. Please check this has been done for all phrases like "good" and "better" are made more quantitative.

*We have attempted to provide more quantitative statements throughout the manuscript as suggested.*

The correction factor of 1.4 for the OH reactivity is interesting (and determined from lab studies of known sinks) – was this seen in the intercomparison also during the ambient measurements?

*The correction factor was determined from additional laboratory studies before and after the peroxy radical intercomparison, and was consistent with the laboratory measurements described in Hansen et al. (2014). This has been clarified on page 9 of the revised manuscript.*

"Laboratory measurements of the reactivity of several VOCs with well-known rate constants, including butane, isoprene, and propane showed that the OH reactivity measurements for these compounds were on average 30% lower than calculated when the measured velocity of the turbulent core is used to determine the reaction time. This consistent underestimation of the OH reactivity is likely due to either incomplete mixing of the reactants or a systematic underestimation of the reaction time, and is similar to that measured previously by Hansen et al. (2014)."

In response to reviewer 2's comment regarding P10 L5, it is stated that "the correlation in the present study was not statistically significant".

Can this be clarified, e.g. in terms of a correlation coefficient r or something similar, otherwise again the statement is a little subjective?

*We have added the R2 value for the correlation as suggested and refined the statement on pages 10-11 of the revised manuscript.*

"This result is also consistent with the measurements of Novelli et al. (2017), who found that their observed interference correlated with the product of ozone and biogenic VOC concentrations, although the correlation in the present study was weak ($R^2 = 0.15$)."

There is a statement in the conclusions about the level of OH before the interference is subtracted (OHwave going up to around 9x10(6)). There ought to be something similar in the abstract. The abstract has the level of OH stated after interference subtracted (4x10(6)) but it is not possible currently to gauge the level of interference from the abstract, and this is something that ought to be there.

*We have added the level of the interference in the abstract as suggested.*

"Using an OH chemical scavenger technique, the study revealed the presence of an interference with the LIF-FAGE measurements of OH that increased with both ambient concentrations of ozone and temperature, with an average daytime maximum equivalent OH concentration of approximately $5\times10^6$ cm$^{-3}$."

The following comments use Page – line number (e.g. 2-17) for the revised MS.

4-25. Bottorff et al., 2015; manuscript in preparation?? Do you mean 2020?

*We have clarified the citations, with the Bottorff et al., 2015 referring to an earlier AGU presentation, and Bottorff et al., 2020 referring to the manuscript in preparation.*

17-9. For the "similar concentrations of OH were observed at this site in 2017..." were these concentrations after interference subtracted - just clarify.

*We have clarified that these concentrations were after the interference was subtracted.*

Figure S8 left hand side panel - shows a budget calculation of OH production, but uses HO2* rather than HO2 to calculate HO2+NO. Could there be a statement (in the caption or the main text) saying what roughly the HO2*/HO2 ratio and hence what the degree of overestimation is of HO2+NO, as there are already missing sources of OH (comparing to the loss rate OHxk(OH)) even using HO2*+NO.

*As discussed above, we have revised this figure using the estimated HO2 concentrations based on the modeled HO2/HO2* ratio. We have indicated the ratio used in the caption to this figure as suggested.*

Reviewer 1

Unfortunately, the revised version is still not satisfactory.

No strong point can be made with the data presented in this paper and therefore there is no advantage of having this study published. The lack of reliable NO data and the inability of the model in reproducing the measured NO when available makes it impossible, in my opinion, to use to data to make any statement on the OH radical production sources and/or missing ones.

*We understand the reviewers concerns, and as discussed above we have confined the modeling to the period when NO measurements were available to provide a more meaningful comparison to the measurements. We have also expanded the discussion of potential missing radical sources based on the results of the models which are now fully constrained by the available NO measurements.*

Also the quality of the radical measurement is questionable. The newly added figure S1 shows the comparison measurements and model for one day with available NO data. First, what can be really learned from having 2 OH data point in 6 hours (between 6 and 12 UTC)? Second, is really the OH radical concentration negative between 14 and 16? What about the huge discrepancy during the night?

*Unfortunately, instrumental problems limited the measurements on the day in question, limiting the number of points for comparing with the model. As discussed above, the large negative values (which occurred primarily at night) reflect the fact that the interference was not measured simultaneously as the OH measurements. Because the interference during these times was much larger than the ambient OH signal, large variations in the interference between measurement cycles often resulted in additional scattering in the ambient OH concentration that was sometimes outside the precision of individual measurements (OH plus interference and the interference)*

*To provide a more robust comparison with the model, we compared the diurnal average of the measurements during the period with NO measurements. The early evening and nighttime measurements during this period are similar to that observed during the entire campaign, and are also similar to measurements made at this site by the IU-FAGE instrument and the University of Colorado CIMS instrument in 2017 as part of an informal instrument intercomparison. The discrepancy between the nighttime measurements and the model suggest that there may be a missing radical source in the model, although additional measurements will be needed to resolve this discrepancy. An expanded discussion regarding the model-measurement agreement has been added on page 12 of the revised manuscript.*

"Including versions of the LIM1 mechanism for HOx regeneration in both the MCM (3.3.1) and RACM2 (RACM2-LIM1) models result in higher modeled daytime concentrations of OH compared to the base MCM 3.2 and RACM2 mechanisms, with the RACM2-LIM1 results in better agreement and within 30% of the measured concentrations during the day (9:00-17:00 EDT) (Fig. 5), while the MCM 3.3.1 model underpredicted the measurements during this period by approximately a factor of 2. These results are similar to that found by Novelli et al. (2020), who found that the MCM 3.3.1 underpredicted measurements of OH by a factor of approximately 1.4 during isoprene oxidation experiments in the SAPHIR chamber. The measured OH concentrations could be reproduced using a model that increased the yield of HPALD in the oxidation mechanism, resulting in an effective bulk isoprene peroxy radical isomerization rate similar to that in the original LIM1 mechanism (Novelli et al., 2020). These larger bulk peroxy radical isomerization rates are incorporated into the RACM2-LIM1 mechanism used in the present study (Tan et al., 2017), leading to the higher modeled radical concentrations compared to the RACM2 and MCM model results shown in Fig. 5."

"Although the RACM2-LIM1 mechanism appears to be able to reproduce the daytime OH radical measurements compared to the MCM 3.3.1 mechanism, all the models underestimate the measurements in the early evening and night. While there is uncertainty associated with these nighttime measurements due to the large interference that was subtracted, similar concentrations of OH were observed during the evening by both the IU-FAGE instrument and the University of Colorado Chemical Ionization Mass Spectrometry (CIMS) instrument at this site in 2017 during an informal instrument intercomparison (Rosales et al., 2018; Reidy et al., 2018). These results suggest that there may be a missing radical source during this period, such as the ozonolysis of unmeasured biogenic VOCs, and additional measurements will be required to resolve this discrepancy."

It is not justifiable to use the HO2* in the budget plot (Fig. S8) saying it is an upper limit. The authors already argue HO2* is affected by the contribution of isoprene RO2 and they even give number to how much those contribute to the total HO2* signal from the model result. So, that contribution should be taken out and only the HO2 radical should be used in the budget as in this case it is not a small interference but the measurement was done to increase the detection of such radicals.

*As discussed above, we chose to use the experimentally measured HO2\* concentrations for the experimental radical budget rather than corrected HO2 concentrations based on model predictions. However, for this revision we have estimated the concentration of HO2 based on the modeled HO2/HO2\* ratio to determine the contribution of the HO2+NO reaction to the experimental OH radical budget and have included other modeled radical sources for comparison in a revised figure.*

To summarize, the quality of the measurements is not such to justify any findings or answers to scientific questions. Indeed there is no strong statement in the paper which just shows data with a comparison with a model but with no effort in making some analysis of the findings as the data do not allow for such. For that reason, I suggest rejection of the paper as publication in ACP requires for a study to have substantial contribution to scientific (substantial new concepts, ideas, methods, or data).

*As mentioned above, we have expanded the discussion of the model-measurement agreement using only the data when all the measurements are available. While this reduces the amount of data, it does provide a more robust analysis of the data, which has been included in the revised manuscript.*

[revised manuscript text omitted]

---

## Author Response (AR3)

Dear Dr. Hofzumahaus,

Thank you for your review of our paper. We have incorporated your suggestions into the revised manuscript. Below are our responses to your comments, highlighted in italics, with changes to the manuscript highlighted in red. The revised manuscript with track changes is attached.

Editor's Comments

The revised paper has been greatly improved and the comments of the reviewers and the editor have been adequately answered. Before the paper can be published, some minor changes are needed for clarification.

Minor changes

The abstract should be more specific and mention which chemical mechanisms have been tested and which version has given the best agreement with the interference corrected OH data. A quantitative statement should be made how much the modeled OH is increased by the LIM1 chemistry. Important boundary conditions (temperature, isoprene and NO concentrations) should be stated.

*We have specified the chemical mechanisms tested in the abstract and have stated that the RACM2-LIM1 mechanism provides the best agreement with the corrected OH measurements. We have also quantified how much the base RACM2 and MCM 3.2 modeled OH concentrations increased with inclusion of versions of the LIM1 mechanism. We have also included the important boundary conditions as suggested:*

This campaign took place in a forested area near the Indiana University, Bloomington campus characterized by high mixing ratios of isoprene (average daily maximum of approximately 4 ppb at 28°C) and low mixing ratios of NO (diurnal average of approximately 170 ppt). Supporting measurements of photolysis rates, VOCs, NOx, and other species were used to constrain a zero-dimensional box model based on the Regional Atmospheric Chemistry Mechanism (RACM2) and the Master Chemical Mechanism (MCM 3.2), including versions of the Leuven isoprene mechanism (LIM1) for HOx regeneration (RACM2-LIM1 and MCM 3.3.1).

The addition of versions of the LIM1 mechanism increased the base RACM2 and MCM 3.2 modeled OH concentrations by approximately 20% and 13% respectively, with the RACM2-LIM1 mechanism providing the best agreement with the measured concentrations, predicting maximum daily OH concentrations to within 30% of the measured concentrations.

Section 2.4: the difference between the two used isoprene mechanisms should be more precisely explained. The implementation in RACM2 as described in Tan et al. (2017) does not include the explicit LIM1 mechanism, but uses bulk reaction rates for the isoprene RO2 isomerization via 1,6-H shift derived as parametrizations from LIM1 by Peters et al. (2014). Contrary, MCM

v3.3.1 contains the full LIM1 mechanism with a description of the equilibrium between different isoprene RO2 isomers and isomerization by H shift reactions of specific isomers (Jenkin et al., 2015). Following a reviewer recommendation by Peeters et al. (2015), Jenkin et al. (2015) adjusted the rate coefficients for equilibration and H shift reactions to match preliminary experimental results by Crounse et al. (2014). Compared to the original LIM1, these changes mean a reduction of the effective bulk rate coefficient for the 1,6-H shift by about a factor of 5.

*We have clarified the differences between the isoprene mechanisms in RACM2 and MCM as suggested on pages 9-10 of the revised manuscript:*

The isoprene oxidation mechanism in RACM2 was updated as described in Tan et al. (2017) to include the Leuven Isoprene Mechanism (LIM1) originally proposed by Peeters, et al. (2009) involving peroxy radical isomerization reactions leading to additional HOx radical production. This is a condensed version of the LIM1 mechanism and includes the updated bulk reaction rate constants for the isoprene peroxy radical 1,6-H shift isomerization reactions as parameterized in Peeters et al. (2014). These isomerization reactions lead to the formation of HO2 and hydroxyperoxy aldehydes (HPALDs) (Crounse et al., 2011; Teng et al., 2017; Berndt et al., 2019) which can subsequently photolyze leading to OH production, as well as di-hydroperoxy carbonyl peroxy radicals (di-HPCARP-RO2) which can rapidly decompose to produce additional OH radicals (Teng et al., 2017; Wennberg et al., 2018). The addition also includes a revision of the chemistry of first-generation isoprene oxidation products, including methyl vinyl ketone (MVK), methacrolein (MACR), and isoprene hydroperoxides (ISHP) (Tan et al., 2017).

In addition, the ambient measurements were also modeled with version 3.3.1 of the Master Chemical Mechanism. In comparison to version 3.2, MCM 3.3.1 incorporates the explicit LIM1 mechanism, including the equilibrium between different isoprene peroxy radical isomers and H-shift isomerization reactions of specific isomers, resulting in HOx radical recycling through the production of HPALDs as well as di-HPCARP-RO2 radicals (Jenkin et al., 2015). Based on the recommendation of Peeters (2015), the equilibrium rate coefficients between different peroxy radical isomers were increased and the 1,6 H-shift isomerization rate constants were decreased in order to match early experimental results of Crounse et al. (2014). These changes resulted in effective bulk 1,6-H shift peroxy radical isomerization rate constants in MCM 3.3.1 that are approximately a factor of 5 lower than the original LIM1 recommended rates (Novelli et al., 2020).

Page 12, line 22-23. Please quantify the difference between the model runs with and without LIM1 chemistry. The corresponding levels of NO from 9 to 17 EDT should be mentioned in this context.

*We have quantified the increase in the modeled OH concentrations due to the LIM1 mechanism as suggested, and have included the average levels of NO from 9 to 17 EDT on page 12 of the revised manuscript:*

Including versions of the LIM1 mechanism in both the MCM (3.3.1) and RACM2 (RACM2-LIM1) models increases the predicted daytime concentrations of OH by approximately 13% and 20%, respectively compared to the base mechanisms during the day (9:00-17:00 EDT), when mixing ratios of NO on these days decreased from an average morning maximum near 500 ppt to approximately 50 ppt in the afternoon.

Page 12, line 22-26. The comparison between the measured and modeled OH concentrations (Figure 5, upper right panel) needs to be done more careful. The statement "... with the RACM2-LIM1 results within 30% of the measured concentrations during the day (9:00-17:00 EDT)" is not true at 14h, where the RACM-LIM1 model and the measurements are numerically different by a factor of 1.8. Probably, this discrepancy is explainable by the total uncertainty of the measured data point (OH calibration error of 18%, plus statistical error bar, plus additional uncertainty from the interference correction mentioned on the same page) and the model uncertainty of 30%.

*We have clarified the agreement between the RACM2-LIM1 model results with the measurements to include a statement that the measurements at 14:00 are still within the combined uncertainty of the measurements and the model on page 12 of the revised manuscript:*

The RACM2-LIM1 results are generally within 30% of the measured concentrations during the day (Fig. 5, top right), with the difference at 14:00 EDT within the combined measurement precision, the accuracy of the calibration (18%, 1σ), and the model uncertainty (30%).

Page 12, line 27-29. The comparison with results by Novelli et al. is only meaningful, if the NO concentrations are comparable. To which NO concentration does the factor of 1.4 refer? The description of the improved mechanism in Novelli et al. is not entirely correct. The improvement is obtained by using the fast rate coefficients for the RO2 equilibrium from MCM v3.3.1 and the fast 1,6-H-shift rate coefficients from the Caltech mechanism (Wennberg et al., 2018). This combination causes a faster production of HPALD and di-HPCARP-RO2, both of which are likely important for the OH regeneration. The overall mechanism has a bulk rate coefficient for the 1,6-H shift which is close to the value in the original LIM1.

*We have included the NO mixing ratio in the Novelli et al. study (less than 0.2 ppb), which is comparable to the NO mixing ratios in this study. We have also clarified the improved isoprene mechanism used in Novelli et al., on pages 12-13 of the revised manuscript as suggested:*

These results are similar to that found by Novelli et al. (2020), who found that the MCM 3.3.1 underpredicted measurements of OH by a factor of approximately 1.4 during isoprene oxidation experiments in the SAPHIR chamber when mixing ratios of NO were less than 0.2 ppb, similar to the mixing ratios of NO measured in this study. The measured OH concentrations could be reproduced using a model that incorporated the larger equilibrium rate coefficients between the different peroxy radical isomers in the MCM 3.3.1 mechanism with the larger 1,6-H shift peroxy radical rate constants from the Caltech isoprene mechanism (Teng et al., 2017; Wennberg et al.,

2018). This combination increased production of HPALDs and di-HPCARP-RO2 radicals in the oxidation mechanism, resulting in an effective bulk isoprene peroxy radical isomerization rate similar to that in the original LIM1 mechanism (Novelli et al., 2020). These larger bulk peroxy radical isomerization rates are similar to that incorporated into the RACM2-LIM1 mechanism used in the present study (Tan et al., 2017), leading to the higher modeled radical concentrations compared to the MCM 3.3.1 model results shown in Fig. 5.

Figure 6 and 8: which measurement days are averaged in the diurnal profiles?

*For Figure 6, we have clarified that the model results are the diurnal average for the days which NO was measured simultaneously, and have specified the days in the main text (page 12).*

*We have also clarified that Figure 8 includes all the measurements made during the campaign.*

Figure 9: what is the meaning of a median diurnal average for a single day?

*We have corrected the caption to indicate that the figure represents the median total OH reactivity measurements for this day.*

Figure 10 and 11: which components are meant by the label 'others' in the legends. Please give a short explanation in the figure captions.

*We have clarified the components of the "Others + hv" categories in the caption of each figure as suggested, in addition to page 16 of the revised manuscript.*

[revised manuscript text omitted]